



# Quantifying uncertainties of satellite $NO_2$ superobservations for data assimilation and model evaluation

Pieter Rijsdijk[1,2, 3,*], Henk Eskes[2,*], Arlene Dingemans[1, ◇], K. Folkert Boersma[2,4], Takashi Sekiya[5], Kazuyuki Miyazaki[6], and Sander Houweling[3,1]

[1]SRON Netherlands Institute for Space Research, Leiden, Netherlands
[2]Satellite Observations department, Royal Netherlands Meteorological Institute, De Bilt, The Netherlands
[3]Department of Earth Sciences, Vrije Universiteit, Amsterdam, the Netherlands
[4]Meteorology and Air Quality group, Wageningen University, Wageningen, The Netherlands
[5]Japan Agency for Marine-Earth Science and Technology, Yokohama, Japan
[6]Jet Propulsion Laboratory/California Institute for Technology, Pasadena, California, USA
[*]These authors contributed equally to this work.
[◇]Currently at Koninklijke luchtmacht

**Correspondence:** Pieter Rijsdijk (P.Rijsdijk@sron.nl)

**Abstract.**

Satellite observations of tropospheric trace gases and aerosols are evolving rapidly. Recently launched instruments provide increasingly higher spatial resolutions with footprint diameters in the range of 2-8 km, with daily global coverage for polar orbiting satellites or hourly observations from geostationary orbit. Often the modelling system has a lower spatial resolution than the satellites used, with a model grid size in the range of 10-100 km. When the resolution mismatch is not properly bridged, the final analysis based on the satellite data may be degraded. Superobservations are averages of individual observations matching the resolution of the model and are functional to reduce the data load on the assimilation system. In this paper, we discuss the construction of superobservations, their kernels and uncertainty estimates. The methodology is applied to nitrogen dioxide tropospheric column measurements of the TROPOMI instrument on the Sentinel-5P satellite. In particular, the construction of realistic uncertainties for the superobservations is non-trivial and crucial to obtaining close to optimal data assimilation results. We present a detailed methodology to account for the representativity error when satellite observations are missing due to e.g. cloudiness. Furthermore, we account for systematic errors in the retrievals leading to error correlations between nearby individual observations contributing to one superobservation. Correlation information is typically missing in the retrieval products where an error estimate is provided for individual observations. The various contributions to the uncertainty are analysed: from the spectral fitting, the estimate of the stratospheric contribution to the column and the air-mass factor. The method is applied to TROPOMI data but can be generalised to other trace gases such as HCHO, CO, $SO_2$ and other instruments such as OMI, GEMS and TEMPO. The superobservations and uncertainties are tested in the ensemble Kalman filter chemical data assimilation system developed by JAMSTEC. These are shown to improve forecasts compared to thinning or compared to assuming fully correlated or uncorrelated uncertainties within the superobservation. The use of realistic superobservations within model comparisons and data assimilation in this way aids the quantification of air pollution distributions, emissions and their impact on climate.



## 1 Introduction

The capabilities of satellite instruments to measure trace gases in the atmosphere have increased greatly in recent years. Instruments measuring from the UV to infrared and microwave (https://earthobservations.org/, https://ceos.org/ourwork/virtual-constellations/acc/), allow the retrieval of concentrations of a large number of gases including $O_3$, $NO_2$, $SO_2$, CO, $CH_4$ and $CO_2$. While a previous generation was providing measurements with footprint diameters of order 15-50 km, instruments like the polar orbiting UV-Vis-NIR-SWIR spectrometer TROPOMI (Veefkind et al., 2012) and the recently launched geostationary instruments GEMS (Kim et al., 2020) and TEMPO (Zoogman et al., 2017) provide observations with a spatial resolution around 5 km, allowing the identification of plumes originating from individual major emitters and the estimation of their emissions (Streets et al., 2013; Georgoulias et al., 2020). At the same time, these instruments provide daily global coverage (TROPOMI) or regional hourly observations (GEMS, TEMPO) resulting in large data volumes (e.g. about half a Tb per day for TROPOMI). Making good use of all this information is a major challenge.

In parallel, global atmospheric composition analysis systems have been developed which use data assimilation techniques to assimilate the available satellite data. In Europe, the Copernicus Atmosphere Monitoring System (Peuch et al., 2022) is assimilating about 24 satellite datasets in real time to constrain concentrations of reactive gases, aerosols and greenhouse gases (Inness et al., 2019b). Multi-decadal reanalyses have been generated by CAMS (Inness et al., 2019a) or by the MOMO-Chem data assimilation system (Miyazaki et al., 2020a).

The recent advances in satellite instruments have led to a mismatch in resolution between models and observations. For example, the TROPOMI instrument has footprints of 5.5 by 3.5 kilometres at nadir (about 20 km$^2$), whereas the Copernicus Atmosphere Monitoring Service (CAMS) model grid cells are roughly 0.4 by 0.4 degrees (about 2,000 km$^2$). As a result, a single model gridcell may be covered by order 100 observations, which will lead to large differences between individual observations and interpolated model values because trace gas concentrations vary strongly linked to the distribution of (point) air pollution sources. Also regional data assimilation or inverse modelling applications, e.g. van der A et al. (2024), often are implemented with a resolution of order 0.2 degree or coarser, with order 10-20 TROPOMI observations per grid cell. This mismatch is the main reason to introduce superobservations, averages of the individual observations which are representative of the scales that are resolved by the model.

Crucial for a successful analysis is high-quality information on both the uncertainties in the model forecast (the error covariance matrix) as well as in the observations. Too optimistic observation errors will lead to spurious impacts in the model degrading the quality of the analyses, while an overestimate of the observation error implies that the observations are not used to their full potential.

The model to data mismatch, or departure $\mathbf{d}$, in equation 1 is a key quantity in data assimilation (Kalnay, 2002). Here $\mathbf{y}$ is the observation vector, and $H$ is the observation operator converting the model state vector $\mathbf{x}$ to the observations.

$$\mathbf{d} = \mathbf{y} - H[\mathbf{x}] \tag{1}$$

There are three sources of error contributing to non-zero $\mathbf{d}$ values: the error in the observation $\mathbf{y}$, the forecast error in $\mathbf{x}$ and the errors in the observation operator, which are often combined with the observation error. The error in $H$ describes how accurately



the measurement can be reconstructed from the model state represented on a finite-resolution grid. This representativity error, although sometimes neglected, will often be the dominant error source. Various terms may contribute to this error, including horizontal spatial representativity errors (Janjić et al., 2018; Schutgens et al., 2016; Miyazaki et al., 2012), temporal errors (Boersma et al., 2016), vertical interpolation errors, smoothing errors (when averaging kernels are not used, see Rodgers (2000))

and forward modelling errors (errors in the radiative transfer model included in $H$ to describe the (satellite) observation). In this paper, we focus on the horizontal spatial representativity error (RE) because this is a major source of error in case of large sub-grid variability and partial coverage. Also, this error is straightforward to simulate and quantify.

In data assimilation applications the uncertainties of the observations are often assumed to be uncorrelated in space because of its complexity. Satellite retrieval products generally contain detailed retrieval error estimates, but these are available for

individual observations and typically there is no information on how much errors in nearby observations may be correlated. If such correlations are neglected the individual observations will strongly impact the analysis. Thinning the observational dataset, using only a subset of the observations often improves the data assimilation results, and reduces correlated errors through data density reduction, while reducing computational cost in data assimilation (Liu and Rabier, 2002, 2003). However, thinning does not decrease the uncorrelated part of the uncertainty (H. Berger, 2003) and leads to a loss of information as well.

An alternative to thinning is 'superobbing'. In this approach, multiple observations are clustered and averaged to a single superobservation. The superobservations then replace the original observations in data assimilation applications as illustrated in figure 1. Various methods of superobbing exist. The clustering of observations inside the optimal interpolation analysis is introduced in Lorenc (1981), but Purser et al. (2000) points out the disadvantages of this method. Another example is the 10s observations constructed for OCO-2 (Crowell et al., 2019). Superobbing methods generally consist of three components: A

method to cluster the observations, to average the observations and to average the uncertainties.

A simple way to cluster observations is by using a pre-determined grid, such as a model grid (Jeuken et al., 1999; Boersma et al., 2016). This minimises the RE between the superobservations and the model. Another approach involves clustering observations based on proximity to a model point in both space and time. Alternatively, clustering observations based on information density can be preferable depending on the desired properties (Duan et al., 2018; Purser, 2015). Detail is retained

where necessary, and more of the structures of the original observations are preserved. This method can retain more information with fewer data points, especially for data with a heterogeneous information density, such as wind data. However, it yields an irregular grid, which may be undesirable. The irregular grid increases the RE between the superobservations and the model.

Various methods exist for averaging clustered observations. The simplest method is to take a mean of all observations part of a superobservation cluster. (Crowell et al., 2019) use the uncertainty of the observation as weights. Miyazaki et al. (2012);

Boersma et al. (2016) average the observations based on their overlap with the superobservation grid.

There are also multiple methods to compute the superobservation uncertainty. Uncertainties may be averaged in the same way as the observations Inness et al. (2019b). On the other hand Crowell et al. (2019) calculate their uncertainty as the largest of square root of the mean variance or the observations standard deviation. H. Berger (2003) and Miyazaki et al. (2012) introduce spatial error correlations between individual observations and combine the uncertainties based on these correlations.

Determining the correlation between the uncertainties can be difficult and can be qualitative (Miyazaki et al., 2012).



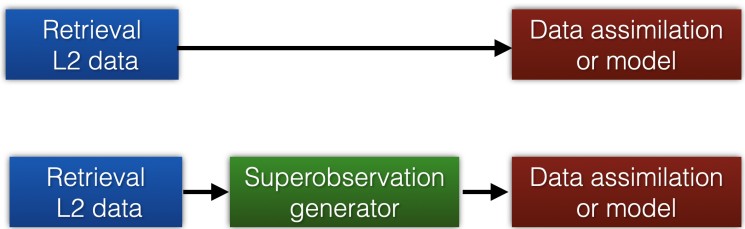

**Figure 1.** The superobservation approach implies that the direct assimilation of level-2 retrievals (top) is replaced by a pre-processing step where a superobservation generator is applied to form an intermediate set of clustered observations with the spatial resolution of the data assimilation system, which is subsequently assimilated or compared with model output (bottom).

The inflation of uncertainties is another method that is often employed to address the problem of correlated uncertainties. Chevallier (2007) demonstrated that inflating observational observational uncertainties gives good results. This method is often combined with thinning to account for the fact that thinned observations are still correlated (Heilliette and Garand, 2015; Bédard and Buehner, 2020). Inflation can also be used in conjunction with superobservations, as superobservation uncertainties are still spatially correlated.

In this paper, we improve and formalize the method used by Miyazaki et al. (2012); Boersma et al. (2016) and apply it to TROPOMI tropospheric $NO_2$ observations for data assimilation applications. The correlations between the retrieval uncertainties are quantified to calculate the superobservation uncertainty more accurately. Also, we derive an equation for the representation error, which has only been parameterized until now. Furthermore, we apply a correction to take into account systematic sampling. We study superobservations with $NO_2$ because it is one of the trace gases most affected by correlated uncertainties and representation errors, due to its short residence time and large variation in both time and space. Also, the high signal-to-noise ratio of the retrieval makes systematic errors dominant over random errors, which makes correctly handling the correlation between uncertainties more important. In particular, we discuss in detail the construction of the superobservation uncertainty, explicitly treating correlations between nearby observations, and the horizontal representativity term, which can partly be described mathematically. In section 2 we give background information on the TROPOMI $NO_2$ product. Section 3 further elaborates on the reasons for making superobservations. Section 4 contains the method we use for superobservation construction and explains the choices for the method. We add to the existing method in sections 5 and 6 by quantifying the correlations between observational uncertainties and the horizontal representativity error. In section 7 we test different methods of constructing the superobservation uncertainty by assimilating the superobservations into the MOMO-Chem data assimilation system.

## 2 Sentinel-5P TROPOMI $NO_2$ observations

The TROPOMI instrument (Veefkind et al., 2012) is a push-broom spectrometer and is the single payload on the Sentinel-5P satellite, which is part of the fleet of Sentinel satellites of the EU Copernicus programme. Four aspects make TROPOMI



unique: 1. The large swath width and resulting daily global coverage; 2. The large spectral range from the UV to the short-

wave infrared, allowing the retrieval of a large number of trace gases like $O_3$, $NO_2$, $SO_2$, HCHO, CO, $CH_4$, as well as aerosol properties; 3. The very high signal-to-noise ratio which allows the retrieval of these gases with a high precision; and 4. The unique small pixels of down to 3.5 x 5.5 $km^2$ at nadir.

The TROPOMI $NO_2$ product data usage and details of the retrieval are provided in the Product Readme File (Eskes and Eichmann, 2022), the Product User Manual (Eskes et al., 2022) and the Algorithm Theoretical Baseline Document (van Geffen

et al., 2022a). The Sentinel-5P Validation Data Analysis Facility (https://s5p-mpc-vdaf.aeronomie.be/) is providing routine validation results with quarterly validation updates.

Two versions of the product are used in this paper. Processor version 2.2.0 became operational in the Summer of 2021, including a new implementation of the cloud retrieval, leading to a substantial increase of the tropospheric columns retrieved (van Geffen et al., 2022b). In combination with high-resolution a-priori information (Douros et al., 2023) this improved the

comparisons to ground-based remote sensing observations (Verhoelst et al., 2021). An intermediate consistent reprocessing of the $NO_2$ data became available on the S5P-PAL server (https://data-portal.s5p-pal.com/products/no2.html). In July 2022 TROPOMI v2.4.0 became operational, including a replacement of the OMI and GOME-2 derived surface albedos in the UV-visible and near-infrared spectral ranged by the TROPOMI directional Lambertian-equivalent reflectivity database (Tilstra, 2023). An official reprocessing of the full mission dataset (30 April 2018 - present) has become available in March 2023. This

most recent upgrade is relevant for this paper because it allows us to study the sensitivity of the tropospheric columns and air-mass factors to uncertainties in the input databases. At the start of this research, the v2.4 reprocessing was not yet publicly available. Instead, we make use of a pre-release processing dataset used for final evaluation. This data is identical to the release data but limited in scope to the first 14 days of September in 2018, 2019, 2020, 2021 and 2022. Thus the analyses in this paper are limited to this timeframe.

A crucial input for estimating the superobservation uncertainty is the error analysis of the individual tropospheric columns. For $NO_2$ the error budget is particularly complex since many aspects contribute significantly to the uncertainty $\sigma_{N_t}$ of the retrieved tropospheric vertical column $N_t$. This follows the error propagation approach developed in Boersma et al. (2004).

$$\sigma_{N_t}^2 = \left(\frac{\partial N_t}{\partial N_{slant}}\right)^2 \sigma_{N_{slant}}^2 + \left(\frac{\partial N_t}{\partial N_{strat}}\right)^2 \sigma_{N_{strat}}^2 + \left(\frac{\partial N_t}{\partial M_t}\right)^2 \sigma_{M_t}^2 \tag{2}$$

The equation distinguishes error contributions from the slant column $N_{slant}$ (uncertainties in the DOAS spectral fit), the

estimate of the stratospheric contribution $N_{strat}$ to the total column, and the uncertainties in the tropospheric air-mass factor $M_t$. The partial derivatives are the error propagation terms or sensitivity of the retrieval to the various sources of uncertainty.

Note that in this equation there is no distinction between random and systematic components of the errors. All terms have quasi-systematic components, e.g. the input surface albedo is available as monthly datasets with a limited spatial resolution which introduces systematic errors, but the satellite sampling of the albedo introduces a random component. For the $NO_2$

slant column retrievals the random and systematic components have been discussed in Zara et al. (2018) and van Geffen et al. (2020).



The tropospheric air-mass factor $M_t$ depends itself on the a priori $NO_2$ profile $x_a$ as well as several input parameters $b$,

$$M_t = M_t(\mathbf{x_a}, \mathbf{b}) = M_t(\mathbf{x_a}, f_c, z_c, a_s), \tag{3}$$

where $f_c$ is the (effective) cloud fraction, $z_c$ is the (effective) cloud height, and $a_s$ is the surface albedo. Note that aerosols are
not treated explicitly in the $NO_2$ retrieval but are implicitly accounted for by the effective cloud fraction and height (Boersma
et al., 2004, 2011).

A basic assumption in the error estimation Eq. 2 is that all terms are uncorrelated. There is one exception: a correlation
term which is introduced between the cloud fraction and albedo. An error in the albedo has a direct impact on the air-mass
factor, but also an indirect impact through the retrieved cloud fraction, partly compensating the direct error (Boersma et al.,
2004). Despite this extra correlation term, the uncertainties in the air-mass factor may be overestimated. In the v2 retrievals,
high biases in the albedo (or LER/DLER) are corrected by matching the observed and computed radiance levels for cloud-free
pixels, which further lowers the impact of the (D)LER input on the final result (van Geffen et al., 2022b). The reduction in
uncertainty due to the albedo adjustment is not accounted for in the v2 uncertainty analysis. The air-mass factor uncertainties
will be further discussed below in section 5.3.

The air-mass factor also depends on the a priori $NO_2$ profile. However, as shown in Eskes and Boersma (2003), relative
comparisons between a model and the $NO_2$ satellite observations become independent of the prior when the averaging kernel
is used in the observation operator. Since the superobservations are constructed for model validation and data assimilation
applications, the kernels should always be applied. Therefore we omit errors related to the a-priori in the remainder of this
study.

The $NO_2$ data product includes averaging kernel vectors $\mathbf{A}$ linking model profiles to the retrieved (tropospheric) columns.
According to optimal estimation theory, these kernels are part of the observation operator and are used to compute a model-
equivalent $y_m$ of the retrieval $y$ by the following equation,

$$y_m = x_a + \mathbf{A}(\mathbf{x} - \mathbf{x}_a). \tag{4}$$

Here $x_a$ is the a priori tropospheric column of the retrieval, $\mathbf{x}$ is the model tropospheric $NO_2$ profile and $\mathbf{x}_a$ is the vertical
profile of layer contributions to the column. Because for the $NO_2$ retrieval (which is to a good approximation linear) we have
$(\mathbf{I} - \mathbf{A})\mathbf{x}_a = 0$, where $\mathbf{I}$ is a vector with elements 1 (Eskes and Boersma, 2003), this reduces to,

$$y_m = \mathbf{A}\mathbf{x}. \tag{5}$$

Note that in this paper, $\mathbf{A}$ refers to the $NO_2$ tropospheric column averaging kernel. These are computed from the total column
averaging kernel by multiplying with the ratio of the total and tropospheric air mass factor (Eskes et al., 2022). Values above
the troposphere as calculated by TM5 are set to 0.



## 3   Why superobservations?

The applications that we have in mind for the superobservations are the assimilation of high-resolution satellite observations with global analysis systems, model validation of global chemistry-transport models or general circulation models including chemistry. Due to its high spatial resolution, there may be many TROPOMI observations in one model grid cell. For instance, for the CAMS global analysis system, this amounts to about 40 to 80 observations in a single model grid cell of about 40x40km$^2$. But even for continental-scale air quality applications (grid cells of 10-20 km) the use of clustered observations may be beneficial.

1. The large number of satellite observations (about one million cloud-free NO$_2$ observations per day) make the assimilation of all observations numerically very costly.

2. Thinning is often used to reduce computational costs and to avoid issues related to spatial correlations between observations. However, in the case of a short-lived tracer like NO$_2$ with local sources the variability within a grid cell of 40x40 km$^2$ is large, and is picked up by TROPOMI. Randomly selecting one observation in a grid cell, or within a correlation length scale, implies throwing away most of the subgrid information and leads to very noisy comparisons because the model does not resolve the fine-scale variability, especially around inhomogeneous point sources. A large representativity error is introduced in this way. Averaging the satellite observations before comparison with the model will imply less noise and reduce the representativity issue.

3. The uncertainty in the individual NO$_2$ observations scales with the column amount (related to uncertainties in the air-mass factor, see previous section). If all individual observations and their uncertainties are assimilated this leads to low-biased analyses. The Kalman gain, $\mathbf{BH}^T(\mathbf{HBH}^T + \mathbf{R})^{-1}$, will give more weight to observations with a small uncertainty (described by the measurement error covariance matrix $\mathbf{R}$) than to observations with a large uncertainty. Given the same uncertainty, low NO$_2$ observations force the assimilation more than high NO$_2$ observations, which yields low-biased results. With the superobservation approach described in this paper, such persistent low biases are largely avoided.

4. Clustering/averaging observations over model gridcells allows us to account for representativeness if the satellite data does not fully cover the model gridcell. The methodology to approximate this error contribution is developed in section 6.

5. The input retrieval product (TROPOMI NO$_2$) does implicitly neglect spatial correlations between retrieval errors of nearby satellite footprints. With the construction of the superobservations, these spatial correlations between observations are taken into account resulting in a single superobservation uncertainty, as discussed in section 5.

6. 14% of the TROPOMI tropspheric NO$_2$ column consists of negative values. This results from the separation of the stratospheric and tropospheric columns. The negative values are necessary to maintain statistical consistency and prevent biases in the retrieval, in particular over remote areas with (almost) no tropospheric NO$_2$. Data assimilation systems are





often unable to use negative values, instead discarding them. This practice results in a positive bias over remote regions. By making superobservations positive and negative values are averaged, which reduces the number of negative values to around 7%. Also on average the negative values less negative, -2.9 $\mu mol\,m^{-2}$ instead of -4.0 $\mu mol\,m^{-2}$. Because superobservations have fewer negative values and the negative values are on average less negative, the bias resulting from not assimilating negative values is strongly reduced.

## 4 Constructing superobservations: the tiling approach

Superobservation construction consists of three components: clustering, averaging and uncertainty averaging. The TROPOMI observations are clustered to the grid of the model the superobservations will be used with, which minimizes the representation error. Additionally, this removes the need for grid interpolation during assimilation. Clustering also significantly reduces the number of TROPOMI observations.

### 4.1 Averaging approach

We average by using the overlap of the individual observations with the grid cell as weights (eq. 6) (Miyazaki et al., 2012; Boersma et al., 2016).

$$y_S = \frac{\sum_i^n w_i y_i}{\sum_i^n w_i} \tag{6}$$

In our formulation, the superobservation is the best possible estimate of the model grid box mean $NO_2$ column given $n$ satellite observations. The weights $w_i$ are obtained by covering (tiling) the grid box with the TROPOMI observations, as shown in Fig. 2. They are equal to the area overlap between the footprint of the TROPOMI observation $y_i$ and the selected model grid box. This method of averaging is similar to spatial binning using the HARP toolbox (http://stcorp.github.io/harp/doc/html/algorithms/regridding.html#spatial-binning). In the rest of this paper normalized weights are used (eq. 7)

$$\tilde{w}_i = \frac{w_i}{\sum_i^n w_i} \tag{7}$$

The tiling method has three main advantages over other averaging methods. Firstly, it takes into account that observations which only partially overlap with the superobservation area should contribute less to the superobservation average. This is especially relevant for smaller superobservations where the difference in overlap becomes more pronounced. Secondly, the tiling method is not sensitive to creating biases. Lastly, the tiling method has a clear physical interpretation, with a closed mass balance. The total amount of tropospheric $NO_2$ in a superobservation is the sum of the tropospheric $NO_2$ of the observations comprising the superobservation. The main alternative of using precision weights assumes every observation within a superobservation is an independent measurement of the superobservation (Taylor, 1997). This is not the case here as different pixels are independent measurements looking at different air masses.





The averaging kernels are averaged in the same way. Multiplying Eq. 1 with $w_i$ and summing over the satellite observations we get,

$$\sum_i^n \tilde{w}_i d_i = \sum_i^n \tilde{w}_i y_i - \sum_i^n \tilde{w}_i \mathbf{A}_i \mathbf{H}_{v,interpol,i}[\mathbf{x}] \tag{8}$$

Here $\mathbf{x}$ is the vector of modelled $NO_2$ partial columns in the vertical layers of the model for the chosen horizontal model grid

box, $\mathbf{A}_i$ is the averaging kernel of observation $y_i$ and $H_{v,interpol,i}$ is the vertical interpolation between the satellite averaging kernel pressure levels and the model pressure levels.

A horizontal interpolation operator is missing because $y_S$ is compared with the model using a single profile of model values for the selected horizontal model grid cell. This is in contrast to an assimilation of individual observations where typically a bi-linear interpolation operator is introduced involving neighbouring horizontal model grid cells.

$$d_S = y_S - \mathbf{A}_S \mathbf{H}_{v,interpol} \mathbf{x} \;;\; \mathbf{A}_S = \sum_i^n \tilde{w}_i \mathbf{A}_i. \tag{9}$$

Thus, the averaging kernel of the superobservation (the "superkernel") is constructed in the same way as the superobservation, using the weights $w_i$. Note that, because all individual observations are by construction compared with the same model value, we do not have to worry about correlations between $\mathbf{A}_i$ and $\mathbf{x}$ (von Clarmann and Glatthor, 2019).

Note that each TROPOMI observation comes with a unique surface pressure, which may differ substantially between neigh-

bouring pixels over mountain terrain. To conserve the total column in the model-satellite comparison, we will follow the TROPOMI $NO_2$ product user manual and align the surface pressures by replacing the retrieval surface pressure with the surface pressure of the model grid cell before comparing. In this way, the kernels of all observations contributing to the super-observation will have the same pressure levels and can be averaged as in equation 9. Note that the shape of the kernel is only weakly dependent on changes in the surface pressure.

## 4.2 Uncertainty averaging

A realistic superobservation uncertainty estimate is essential to guide the data assimilation and to find the right balance between the model forecast and the observations in the analysis. The total uncertainty of the superobservation $\sigma_S$ is the combination of the measurement error and representativity error terms, assuming these are uncorrelated (eq. 10).

$$\sigma_S = \sqrt{\sigma_{obs}^2 + \sigma_{RE}^2}. \tag{10}$$

The observational uncertainty of the superobservation depends on the uncertainty of the individual observations as well as their correlation. To calculate the former, we apply the method from Sekiya et al. (2022) who calculate the retrieval contribution to the superobservation uncertainty using equation 11. This assumes a representative uniform correlation factor c, which is applied to all uncertainties within a superobservation. Here the observational uncertainty is a combination of an uncorrelated





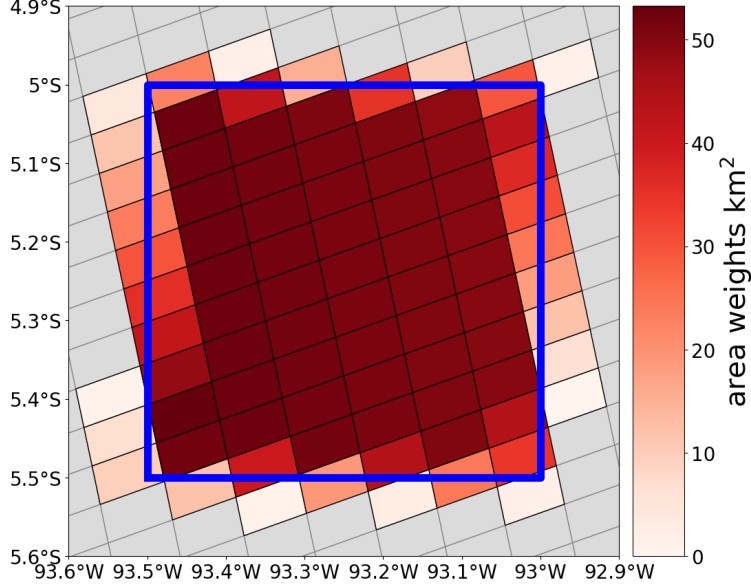

**Figure 2.** The tiling approach: a model gridcell mean $NO_2$ tropospheric column amount is constructed as an overlap area-weighted average of the satellite footprints covering the model grid cell. The colours indicate the weights $w_i$ (unit $km^2$). The grid cell boundary is indicated in blue.

part and a correlated part. The uncorrelated part tends towards zero as the number of observations increases because the square

of the standardized weights $\tilde{w}_i^2$ decreases. On the other hand, the correlated part does not change much when adding more observations. As a result, the correlated part puts a lower limit on the uncertainty, which is roughly $\sigma\sqrt{c}$.

$$\sigma_{obs}^2 = (1-c)\sum_{i=1}^{N}\tilde{w}_i^2\sigma_i^2 + c\left(\sum_{i=1}^{N}\tilde{w}_i\sigma_i\right)^2 \tag{11}$$

## 5 Uncertainty estimate for TROPOMI $NO_2$

As mentioned in section 4.2, the superobservation uncertainty is calculated using equation 11. If the uncertainties within the

superobservation are equal and fully uncorrelated, the uncertainty reduces to $\frac{\sigma}{\sqrt{n}}$. When fully correlated, the superobservation uncertainty becomes the (weighted) average uncertainty.

The spatial correlation will be different for each of the components contributing to the total uncertainty of the observation errors, see equation 2. The correlations for each component separately are discussed below.





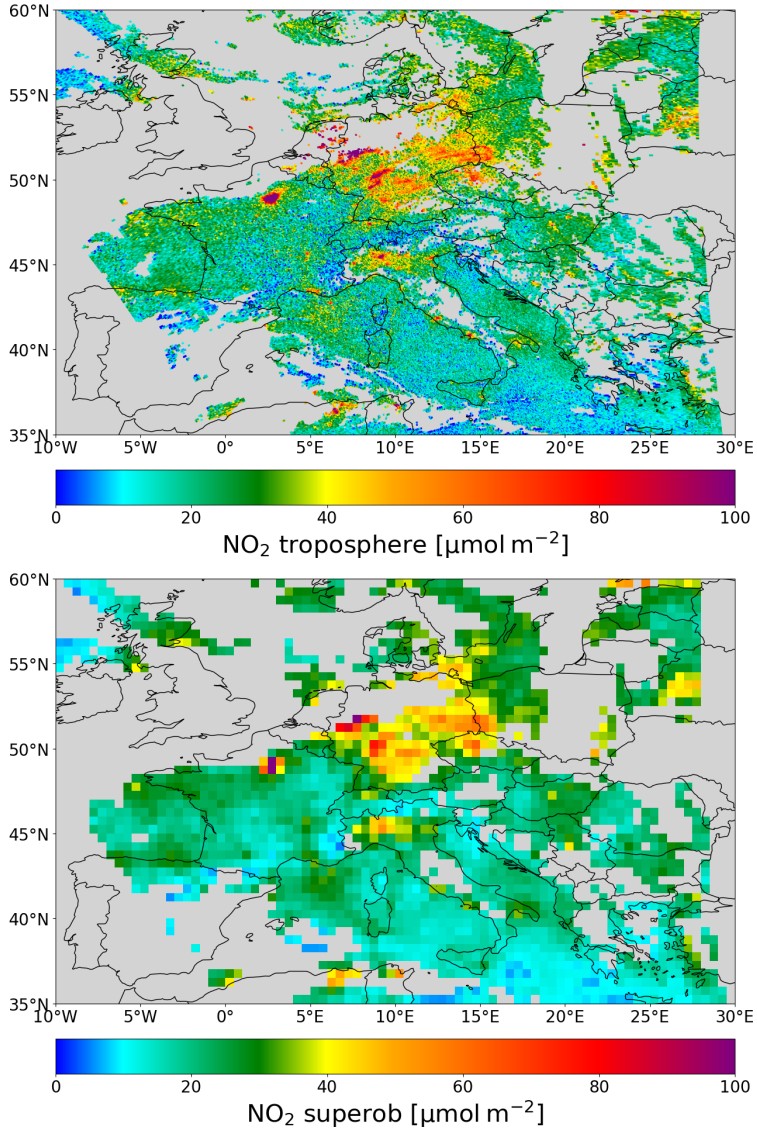

**Figure 3.** An example orbit of TROPOMI on 2018-09-08 $NO_2$ over Europe (top) and the corresponding superobservations (bottom) for a model grid of 0.5x0.5 degrees constructed with the tiling approach. Cloud-covered observations have been filtered out by using only observations with a qa-value > 0.75. Locations without data are coloured grey.

## 5.1 Stratospheric uncertainty

The tropospheric $NO_2$ column is obtained by subtracting an estimated stratospheric column from the total observed column. The stratospheric column is obtained by TM5-MP model simulations while assimilating TROPOMI $NO_2$ column observations (Huijnen et al., 2010; Dirksen et al., 2011). With the method, the stratospheric column is constrained by the TROPOMI





observations, with strong forcings in the assimilation over unpolluted areas, such as the oceans, and small adjustments over polluted regions. Subtracting the modelled stratospheric slant column from the total slant column and dividing the tropospheric

air mass factor gives tropospheric $NO_2$ (eq 12).

$$N_{trop} = \frac{N_{slant} - N_{s,strat}}{M_{trop}} \tag{12}$$

The resolution of the TM5-MP model is 1x1 degree, and the horizontal correlation length scale used in the assimilation is about 500km, both coarser than the superobservations sizes considered in this paper. Therefore, the stratospheric uncertainty is assumed to be fully correlated between observations that are part of one superobservation.

Because fully correlated terms will influence the final superobservation error stronger than uncorrelated or partially correlated terms, the stratospheric estimate will become relatively more important compared to other sources of uncertainty. Therefore it is relevant to investigate this term in more detail.

There is seasonal and latitudinal variation in the stratospheric uncertainty. However, the TROPOMI $NO_2$ retrieval approximates the stratospheric uncertainty using a constant mean value. To improve on this, we analyse the observation - forecast

(O-F) departure between the TROPOMI and model column, using a geometric air-mass factor for both (eq. 13). The RMSE is calculated daily over 5-degree latitudinal bands, highlighting latitudinal and temporal uncertainties. Only areas with an average model-estimated tropospheric $NO_2$ column lower than 30 $\mu mol\,m^{-2}$ are included to minimize the effect of the troposphere. Figure 4 shows clear latitudinal and seasonal variations of the TROPOMI and TM5 differences. To reduce noise in the data a block function convolution is applied to smooth the data over 15 degrees and two weeks. The smoothed data is oversampled

into bins of two degrees by one day. To calculate the geometric stratospheric uncertainty $\sigma_{Stratgeo}$ for an observation this data is linearly interpolated to its day and latitude. If an observation occurs outside of the bounds of the data it is set to the maximum of the data. These gaps result from the lack of observations during polar nights. Equation 14 converts the geometric stratospheric RMSE to the stratospheric uncertainty (van Geffen et al., 2022a).

$$M_{geo} = \frac{1}{\cos\Theta_0} + \frac{1}{\cos\Theta} \tag{13}$$

$$\sigma_{strat} = \frac{\sigma_{Stratgeo} \times M_{geo}}{M_{trop}} \tag{14}$$

In general, areas closer to the poles have a higher RMSE. This is more pronounced in the northern hemisphere because the higher $NO_2$ concentrations in the northern hemisphere increase the absolute errors. In winter the polar region is not observed, and model biases will build up, affecting concentrations in late winter. Also, there is seasonal variation in the high latitudes which relates to the formation and breaking of the polar vortex during winter, leading to larger errors. Gradients around the

Antarctic vortex are also challenging to predict, particularly during souther-hemisphere spring. Because the Antarctic vortex is more stable, these errors are less pronounced and occur during the southern hemisphere spring. High latitude summer $NO_2$





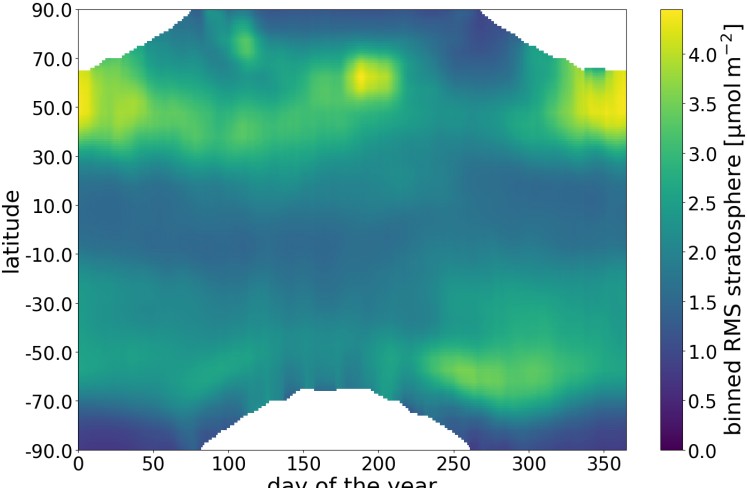

**Figure 4.** Zonal average of the observation - forecast root mean square error of the TROPOMI total column - TM5 total column, averaged over multiple years (2018-2022). Data is smoothed over 15 degrees and two weeks and averaged into bins of one degree by one day. This result is used as the geometric stratospheric uncertainty $\sigma_{Stratgeo}$ instead of the constant in the retrieval.

levels are also difficult to predict. This relates to Arctic fire emissions from Siberia and Alaska. In TM5-MP these are based on climatological fire intensities from GFED, meaning the model is not capable of accurately predicting individual fire events and the corresponding total and stratospheric column. In contrast, in the tropics, the RMS results are better than the mean value

because of the relatively small natural variability there. Compared to the constant $\sigma_{Stratgeo}$ of 3.32 µmol m$^{-2}$ part of the data product the new uncertainty is generally lower, especially at the equator. Depending on the season the uncertainties can become higher in higher latitude areas. This includes some high NO$_2$ areas such as Europe and the United States.

### 5.2 Slant column

Measurement noise is contributing to the slant column uncertainty. van Geffen et al. (2020) finds an average random slant

column uncertainty of 10.23 µmol m$^{-2}$ for cloud-free scenes. Apart from a random component to the slant column uncertainty, there will also be a (regional) systematic component. The systematic component consists of gaps in knowledge, such as missing cross sections, inaccurate Ring coefficients in the DOAS fit or the lack of an intensity offset and a correction for vibrational Raman scattering. (Richter et al., 2011). These systematic effects are most pronounced over the sea in clear-sky conditions. In such circumstances, the systematic uncertainty can be larger than the random uncertainty. But because these are low NO$_2$

environments the impact on the retrieval is limited.

This regionally systematic component is present in the slant column and influences the observation - forecast between the TROPOMI and model geometric column. A systematic error in the slant column results in an increase in the RMSE of the O-F, assuming the model does not exhibit the same systematic error. Moreover, the assimilation and transport of the systematic error within the model results in a further increase in the (O-F) RMSE. Considering that the effect of the systematic error is




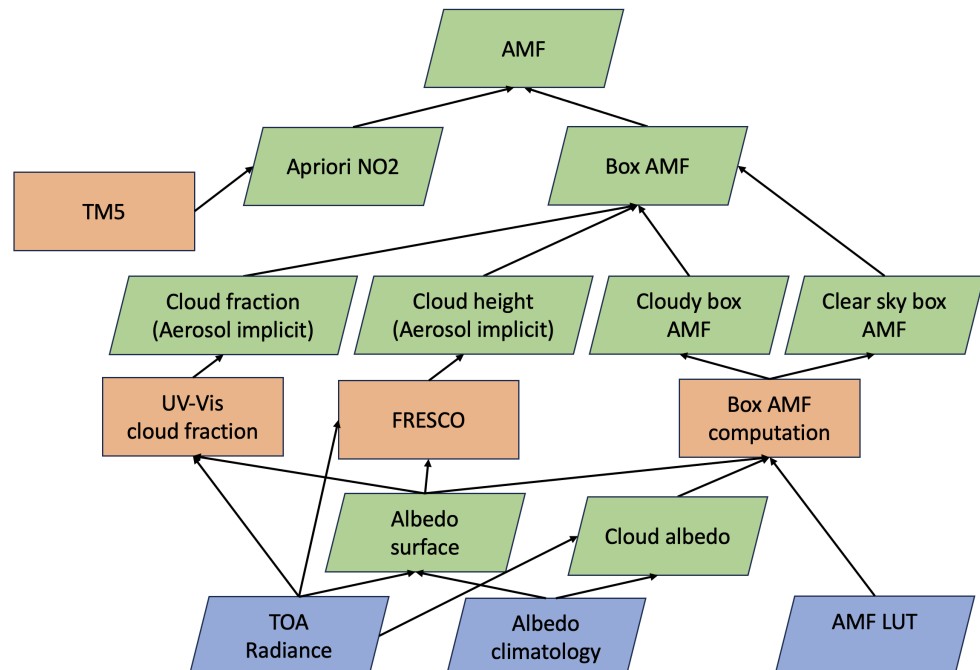

**Figure 5.** Diagram showing the dependencies (arrows) in the calculation of the air-mass factor (AMF) as part of the TROPOMI NO$_2$ tropospheric column retrieval. Shown in blue are the input data (TROPOMI radiances, albedo climatology and static data), in orange are the processing blocks (cloud properties and box AMF lookup table evaluation, TM5 CTM) and in green are the (intermediate) products.

already (partially) included in the stratospheric (O-F) RMS discussed above, we do not separately quantify the effect of the systematic retrieval error. Instead, only the random part of the slant column uncertainty from the level two data is converted to a tropospheric column uncertainty through the tropospheric AMF and averaged as uncorrelated using equation 11.

**5.3 Air mass factor uncertainty**

For the uncertainty resulting from the calculation of the air mass factor the uncertainty of individual observations that is
computed in the level 2 retrieval is used. Calculating the associated spatial correlation between observations is not trivial because the tropospheric air-mass factor $M_t$ is calculated through several inputs, algorithms, dependencies and feedbacks, as shown in figure 5. One of these complicating factors is the use that is made of the top-of-atmosphere (TOA) radiance to correct the albedo climatology for dark scenes. Uncertainties in the algorithms and input variables induce uncertainty in the AMF. Of these uncertainties, the a priori NO$_2$ profile is a large contribution, typically ranging from 5-20% in polluted regions.
These are most affected because the low resolution of the a-priori profile may result in the underestimation of hotspots (Douros et al., 2023). However, as shown in Eskes and Boersma (2003), relative comparisons between a model and the NO$_2$ satellite observations become independent of the a priori profile shape when the averaging kernel is used in the observation operator.





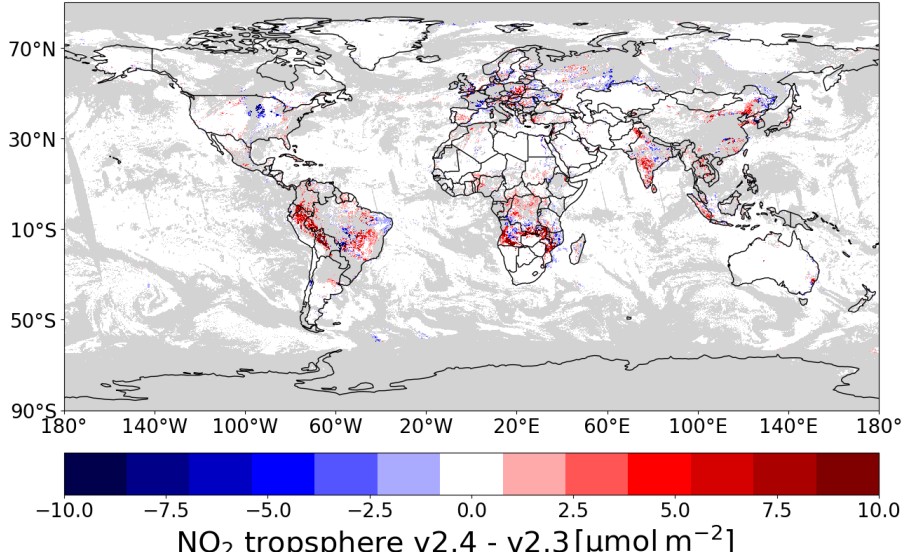

**Figure 6.** Difference between the tropospheric column of the v2.3.1 and v2.4 products on 14-09-2018. Data is filtered for QA > 0.75 (grey areas indicate cloud cover). White colours range from -8 to 8. $\mu$mol m$^{-2}$

Since the superobservations are constructed for model validation and data assimilation applications, the kernels should always be applied. Therefore we omit errors related to the a priori in the discussion below.

Other large sources of uncertainty are the effective cloud cover, the effective cloud height and the surface albedo (or the Lambertian Equivalent Reflectivity, LER). Aerosols are treated implicitly through the effective cloud fractions and cloud height, which introduces a minor uncertainty (Boersma et al., 2004, 2011). All three of these variables depend on a climatological surface albedo dataset. For S5P-PAL NO$_2$ processor version 2.3.1 these are derived from OMI (440nm) and GOME-2 (758nm) while for v2.4.0 it is derived from TROPOMI spectra. A typical RMS difference between these two albedo datasets at 440nm is

0.015, or about 25% for a typical albedo of 0.06. Furthermore, the uncertainties are spatially correlated, first of all, because of the relatively low resolution of the LER database, but also because surface modifying conditions are often spatially extensive. For example, droughts impact the surface albedo in a large area. Luckily the retrieval algorithm can partially compensate for errors in the climatological surface albedo. If the TOA radiance is lower than expected based on the albedo, the albedo is adjusted downwards. On the other hand, if the TOA radiance is higher than expected it is attributed to "effective" clouds. If

these clouds are placed at the correct heights (e.g. at the surface for a high albedo anomaly), this yields approximately the same AMF as with a perfect surface albedo (Riess et al., 2022).

To estimate the spatial correlation required to estimate the superobservation uncertainty we compare versions 2.3.1 and 2.4 of the retrieval. We use the data of the first two weeks of September for 2018-2022 because these were processed as validation before the product was made publically available. The difference between the datasets should be representative of

the uncertainty resulting from the climatological surface albedo, as both datasets are valid inputs. Albedo is also a key input





for the cloud retrieval, so this replacement also generates differences in cloud fraction and cloud pressure. One may argue that the comparison is not a good estimate of the uncertainty in v2.4, because the new TROPOMI surface albedo is likely superior. Thus the uncertainties obtained here are overestimated. However, both maps are still climatological. This absence of temporally explicit data is probably a major source of uncertainty, making both maps uncertain.

Figure 6 shows the difference in $NO_2$ on 14-09-2018. These differences, caused by the replacement of the albedo climatology, are spatially correlated. A correlation length is calculated as outlined in appendix A. This correlation length is then used to calculate the average correlation of a superobservation for use with equation 11. Using a correlation length is preferable over using an average correlation because it takes into account that high-latitude superobservations have a smaller surface area than low-latitude ones and thus should have a higher average correlation if other factors are equal. Also, a correlation length is

resolution agnostic, which allows for an easy change of the superobservation resolution and a properly behaved limit towards smaller superobservations. The correlation $C$ between two points at a distance d for a correlation length $l$ is calculated using the exponential form, $C = exp(-d/l)$

    We calculate the correlation for every distance within a superobservation and multiply this with the probability density function of points within a box (Philip, 1991). Integrating this yields the average correlation within a superobservation. Note

that strictly speaking the PDF from Philip (1991) is for a cartesian plane, not for a sphere, but grid cells are rectangular to a good approximation except very close to the pole.

    The difference between the versions is compared to the uncertainty due to the AMF, as estimated by the retrieval. The RMSE is calculated per swath in a 1-by-1-degree grid and then averaged. The uncertainty is averaged to the same grid. Figure 7 shows the relationship between these variables. There is a relationship between them with an R-value of 0.724. On average

the uncertainty estimated by the retrieval is higher than the RMSE, with a slope of 0.747 of the Theil-sen estimator. Note that factors other than the surface albedo contribute to the uncertainty, such as the choice of radiative transfer model, the wavelength at which the AMF is calculated, sphericity corrections and systematic aspects in the cloud retrieval (Lorente et al., 2017). Based on this information the difference between v2.4 and v2.3 is consistent with the retrieval uncertainty. Thus, the obtained correlation length is likely representative of AMF uncertainty.

**6   Representivity error**

For the superobservations, the data are clustered based on a pre-defined grid. However, often observations will not completely cover a superobservation grid cell, mainly because of cloud cover. Averaging the available measurements only approximates the true mean over the superobservation grid cell. The difference between the true and estimated mean is the horizontal representativity error which is estimated in this section.





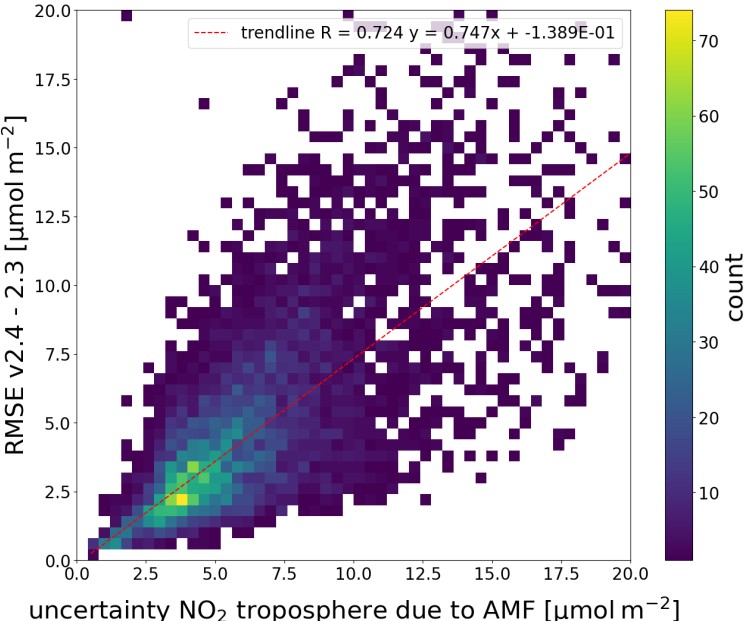

**Figure 7.** Binned scatter plot between the retrieved average tropospheric uncertainty resulting from the AMF and the average RMSE of the difference between the retrieval versions. Calculated using available data globally. Trendline fit using a Theil-sen estimator.

## 6.1 Representativity error due to a random removal of observations

The representativity error (RE) can be quantified by comparing the mean of a completely covered grid cell to the mean of several samples taken from that grid cell. Which is given by:

$$RE = |\mu - \bar{x}_n| \tag{15}$$

$\mu$ is the true mean of the fully covered grid cell and $\bar{x}_n$ is the sample mean based on a subset of $n$ observations. This can be computed by repeatedly sampling the grid cell in a random way. We start by sampling a single point, which we use to estimate the mean of the grid cell. Then we repeatedly add random points and estimate the mean using the available points. Because the order in which the points are sampled influences the result, we perform multiple iterations of this process. The uncertainty of the mean is the standard deviation on the estimate of the mean, $\sigma_{RE}$ (eq. 16).

$$\sigma_{RE,n} = \sqrt{\frac{1}{I} \sum_{i=1}^{i=I} (\mu - \bar{x}_n^i)^2} \tag{16}$$

$I$ is the number of iterations for computing the standard deviation on the mean, and $n$ is the number of samples used to estimate the mean. Figure 8, top panel, shows the result of equation 16. The thin lines are individual iterations, and the green line is the standard deviation of the difference from the mean. To enable comparisons between grid cells and the theoretical method of calculating the RE, the y-axis is normalized by the standard deviation of a cell.





**Figure 8.** Results of repeatedly sampling a single grid cell to calculate $\sigma_{RE}$. (a) random sampling of a single superobservation. The thin grey lines represent individual random experiments for the superobservation. The green line is the mean of the samples, and the blue line is the theoretical result from equation 17 (b) example of a random sample at 30% coverage (c) Systematic sampling of a single superobservation. The red line is the mean of the samples, and the blue line is the theoretical result for the random case. The purple line shows the fit to the systematic mean by fitting $N_{eff}$. In this case the fitted $N_{eff}$ is 5.5 compared to 536 observations. (d) Example of a systematic sample for 30% coverage.



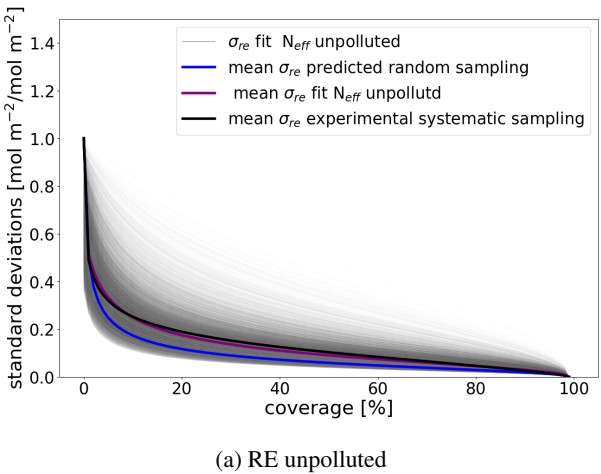
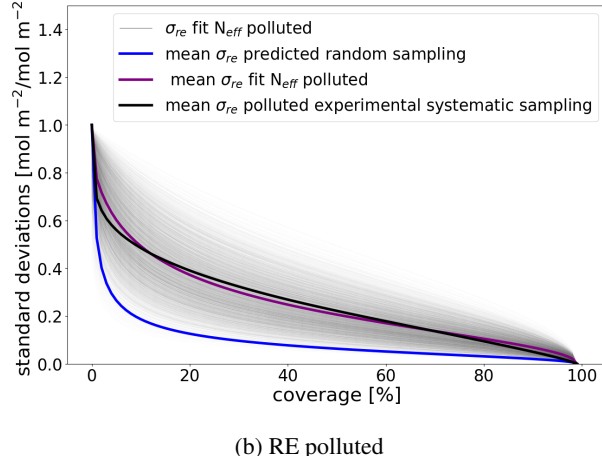

(a) RE unpolluted

(b) RE polluted

**Figure 9.** Result of systematically sampling multiple cells. The thin grey lines are the fit of a single cell using the number of effective observations, similar to the purple line from figure 8. The purple lines show the average fit for unpolluted cells(a) and polluted cells(b). The blue lines are the predicted averages over multiple superobservations using equation 17. The black lines are the averages over multiple superobservations when sampling systematically, for unpolluted cells in a and polluted cells in b.

A mathematical parallel for the experimental results has been found. In essence, $\sigma_{RE}$ is the standard deviation of the sample mean to the true mean. By definition, the standard deviation of the sample mean is the standard error. To calculate $\sigma_{RE}$ the method for calculating the standard error is used. Because the number of observations within a superobservation is finite and we sample without replacement, a correction factor is applied (Bondy and Zlot, 1976; Isserlis, 1918). The corrected formula for the standard error is shown in 17, for sample size $n$, population size $N$ and the standard deviation $\sigma$ of the observations within the superobservation.

$$\sigma_{RE,n} = SE = \frac{\sigma}{\sqrt{n}} \sqrt{\frac{N-n}{N-1}} \tag{17}$$

The results from this equation are shown as the blue line in figure 8. Like the experimental calculation, it is normalized by the grid cell standard deviation. The results from the experimental and theoretical methods are almost identical, showing that formula 17 is a suitable method for quantifying $\sigma_{RE}$ for random sampling. For an unweighted average, the experimental results converge to the theoretical solution. Because the superobservation average is area-weighted, there is a slight difference between the two, which is more pronounced for higher-resolution superobservations because they have a larger variance in the weights. Note that the case where a superobservation is covered by only one observation is analogous to thinning. In this limit, the uncertainty is increased by the standard deviation of the observations in the model grid (n=1 in equation 17).

In the context of the superobservations, many observations at the edges only partially overlap the grid cell (Fig. 2 with reduced weights, so we need to define what we mean by a single observation. Should the input for equation 17 be based on the number of unique observations contributing to a superobservation? In this case, a superobservation with a lot of partially





overlapping observations would have a lower RE, compared to a super observation with the same coverage but fewer observations. This is not desirable because in both cases there is roughly the same amount of information on the superobservation. If anything, the case with a lot of partially overlapping observations is less representative because the partially overlapping observations are influenced by concentrations from outside of the superobservation. Instead, we use fractional observations, 

meaning that if only half of an observation is within a superobservation, it only counts as half an observation. To account for the fact that observations are of a different size we normalize the count by area. The population size $N$ is calculated as the sum of $w_i$ divided by the overlap weighted mean satellite footprint area. $n$ is then calculated as the population size $N$ multiplied by the fractional coverage $f$, which is the fraction of the superobservation covered by valid observations. This method also works well for smaller superobservations because the fractional number of observations is not sensitive to the exact placement

of the superobservation, whereas the unique number of observations with a non-zero overlap is sensitive to the placement of the superobservation.

Another kind of representation error is caused by the fact that observations only partially overlapping with the superobservations are less representative. Smaller superobservations are impacted more by this because they consist of more of these observations. To quantify this one would need sub-TROPOMI scale information which is outside of the scope of this research.

When making superobservations based on just a few observations one should be mindful of this effect.

### 6.2    Representativity error due to a systematic removal of observations

A major complicating factor is that the coverage of the superobservation is not random. A cloud field could cover the northern half of the superobservation. The valid observations then only cover the southern half of the superobservation making it less representative of the grid cell as a whole than a random sample. We repeat the experiment from section 6.1 but instead, sample

systematically. The systematic sampling of a grid cell starts by picking a random observation from the grid cell. Then the nearest observation is added to the sample, which is repeated until the grid cell is filled. This is done for multiple iterations, resulting in figure 8, bottom panel, with the iterations in grey and the experimental $\sigma_{RE}$ in red. As expected, the systematic experiment produces a higher representativity error. At 50% coverage, the increase in RE is 54% for clean areas and 263% for polluted areas.

This increase in RE is parameterized by fitting the total population size $N$. By lowering the population size $\sigma_{RE}$ increases for the same fractional coverage. In this way, we can introduce an effective population size $N_{eff} < N$ such that we match the curve obtained for systematic sampling. The effective number of observations $n$ for that effective population size $N$ is calculated as $N$ multiplied with the fractional coverage $f$

Equation 17 is modified to equation B4 for fitting the population size of a superobservation. This is elaborated in appendix

B. Fitting equation B4 results in the purple line in figure 8, which is fit to the systematic sampling experiment.

$$\sigma_{RE,n} = \frac{\sigma}{\sqrt{N_{eff}f + 1}}\sqrt{1 - f} \tag{18}$$





The effective population size of a superobservation has a physical interpretation. Imagine a superobservation containing two distinct regions: a city with high tropospheric $NO_2$ levels and a rural area outside the city with low tropospheric $NO_2$. If we were to sample the entire city (including the pollution plume from the city), the estimate of the superobservation average is not

much better than with a single sample over the city. Effectively, there are only two independent observations, the city and the rural area. As the example illustrates, the effective population size of observations in a superobservation depends on its spatial structure. If the effective population size is the same as the regular population size there is no effect of systematic sampling on the superobservation. This occurs over areas such as the oceans and the Sahara, where observed tropospheric $NO_2$ is similar and noise-dominated. If the values within the superobservation are random, systematic sampling has no effect. On the other

hand, source regions are sensitive to systematic sampling and applying it gives very different results. Major population centres, such as China, the Middle East and Europe, all have a significantly lower effective population size than the actual population size. Regions with fire emissions, such as the savannahs in Africa, are also sensitive to systematic sampling. The effective population is a property of a location and can be quantified for that location.

To calculate a representative effective population size for a location, the $sigma$ normalized RE on that location is calculated

and averaged over the dataset. The average is used to fit an effective population for that location, which we compare to the average population size for the location. The ratio $R_{eff}$ between the time-averaged population size $\langle N \rangle$ and the effective population size $N_{eff}$ captures how sensitive that location is to systematic sampling ($R_{eff} = \langle N \rangle / N_{eff}$). This ratio $R_{eff}$ is used to calculate $N_{eff}$ for not fully covered superobservations,

$$N_{eff} = N/R_{eff}. \tag{19}$$

While it is possible to calculate $R_{eff}$ for every superobservation, this quantification would be grid-dependent, which would make the method very inflexible. Instead, we calculate an average ratio $R_{eff}$ for polluted superobservations and unpolluted superobservations at multiple resolutions. As demonstrated by our previous example, having emissions sources within or in proximity to a superobservation makes it more susceptible to systematic sampling. To capture this effect we distinguish between polluted and unpolluted superobservations. Superobservations over 30 $\mu$mol m$^{-2}$ are classified as polluted and sensitive to

systematic sampling. In Fig. 9 we calculated $R_{eff}$ for polluted and unpolluted superobservations. First, we calculate $R_{eff}$ for individual locations, by fitting $N_{eff}$. The fits are shown as the thin lines in Fig. 9. This is done separately for times when the locations are polluted or unpolluted. Note there are fewer fits for the polluted case because many locations are never polluted. Then we average $R_{eff}$ to obtain the average $R_{eff}$ of 21 and 3 for polluted and unpolluted 1-degree superobservation respectively. The purple line in Fig. 9 shows the average result of the fits, which matches well with the average experimental

value in black.

Another important factor that influences how sensitive a superobservation is to systematic sampling is the area of the super-observation. Increasing the area of a super-observation increases the mean distance between observations within a superobservation, which makes it more likely to miss information when systematically sampling the superobservation. Figure 10 shows the $R_{eff}$ as a function of the superobservation area, which increases as the area increases for both polluted and unpolluted su-





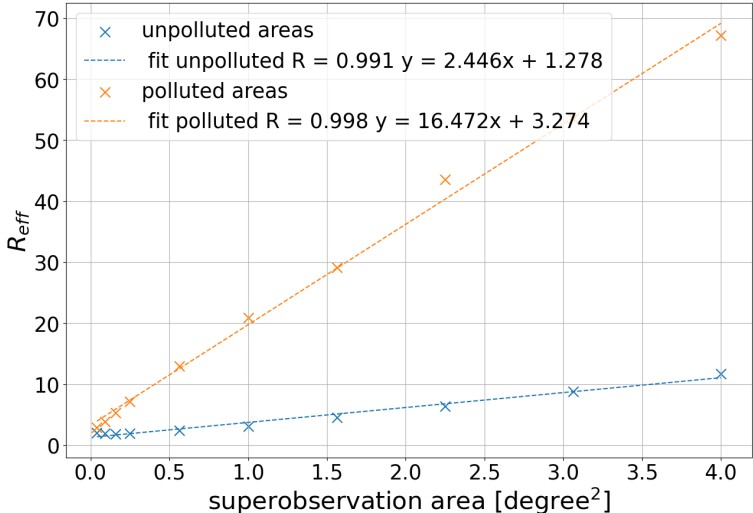

**Figure 10.** Resolution dependency of the correction for systematic sampling, as a function of the area of the superobservation up to a resolution of two degrees. The plot distinguishes between polluted ( $>30\,\mathrm{\mu mol\,m^{-2}}$; orange curve) and unpolluted (blue curve) grid boxes.

perobservations. Within our software, we use the trendlines in Fig. 10 to quantify the $N_{eff}$ of a superobservation. This allows us to calculate $N_{eff}$ for different grids and to take into account latitudinal variations in grid-cell size.

### 6.3   Sample standard deviation

Thus far, the RE has been expressed in terms of the observation standard deviation within the grid cell. This standard deviation may be derived from a climatological, resolution-dependent pre-computed map, or may be estimated using the measurement

variability within each superobservation. We implemented the latter approach. To reduce the number of unrealistic standard deviation estimates, we recommend using a coverage of at least 30%. This coverage corresponds to the point where on average the sample standard deviation would be more accurate than a climatological standard deviation (determined for a 0.5-degree superobservation size). With this coverage, it is still possible that there are not enough available data points to calculate a reliable standard deviation, in particular for smaller superobservations. An alternative approach for smaller superobservations could be

to set the minimum coverage to 50% or even 70%. When fewer than 5 data points are available we set the standard deviation equal to 0.4 times the tropospheric column + $2.5\,\mathrm{\mu mol\,m^{-2}}$. This is based on the relationship between the tropospheric column and the standard deviation(appendix C). Unrealistic standard deviations can occur by chance for individual superobservations. To mitigate this we introduce two lower bounds: Firstly a lower bound of 0.25 times the tropospheric column. This corresponds with the lower bound from C1. Secondly a lower bound of $2.5\,\mathrm{\mu mol\,m^{-2}}$, which corresponds to the intersection of the trendline

with the y-axis.



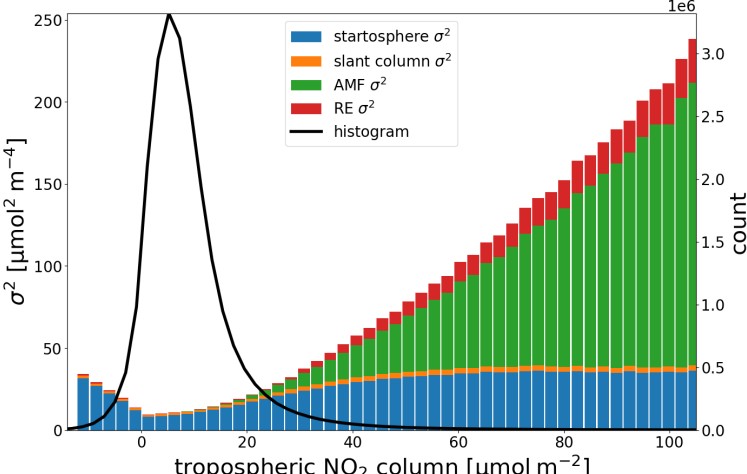

**Figure 11.** The division of the superobservation error variance into its components: stratosphere (blue), slant column (orange), air mass factor (green) and representivity (red). Computed from TROPOMI $NO_2$ at a 0.5 degree resolution. Note that the figure depicts the error variance uncertainty instead of the uncertainty because the variance is a direct sum of its contributions. The black line shows the number of observations within each column bin.

## 7   Combined uncertainty of the superobservations

Figure 11 shows the contributions to the superobservation uncertainty[2] as a function of the tropospheric $NO_2$ column. For low tropospheric columns the uncertainty is dominated by the stratospheric uncertainty, while for high tropospheric columns, it is impacted most by the air mass factor with still a major contribution from the stratospheric uncertainty. The RE is only a minor
contribution to the average uncertainty, but it varies significantly by location depending on the coverage and standard deviation (as illustrated in Fig. 13c below) and becomes important at the edges of cloud fields. The slant column uncertainty has almost no impact on the average uncertainty, even though it is a major source of uncertainty for individual observations over clean areas. Because the slant column uncertainty is treated as dominantly uncorrelated it is reduced significantly by the averaging process. Note that the systematic slant column uncertainty is (partly) included in the stratospheric uncertainty.

We constructed and created superobservations by combining all sources of uncertainty, as described in sections 4.2 and 6. Figure 12a shows the constructed tropospheric $NO_2$ superobservations on a grid of 0.5x0.5 degree$^2$ for the overpass on 8 September 2018. Additionally, Fig. 12 shows superobservations created using weights determined from the uncertainty of the individual observations ($w_i = 1/\sigma_i^2$) (Fig. 12c) and by using random observations (analogous to thinning), Fig. 12d. For comparison, the satellite observations have also been included Fig. 12b). The regular superobservations and the uncertainty
superobservations are similar. Both give a realistic low-resolution representation of the original satellite data. But, as expected, the uncertainty-weighted superobservations have systematically lower values because the weights favour the smaller columns. This is most clearly observed above Paris and North Africa. On average the uncertainty weighted superobservation in Figure 12 have a tropospheric column of 22.4 $\mu mol\,m^{-2}$, compared to 23.0 $\mu mol\,m^{-2}$ of the normal superobservations, which is a







**Figure 12.** Panel showing various methods of pre-processing observations for data assimilation on 2018-09-08 for qa > 0.75. (a) superobservations constructed for this research (b) regular TROPOMI observations (c) uncertainty-weighted superobservations, instead of the area weights used by this research (d) Random sample from the observations within the model grid.





**Figure 13.** Panel showing various methods of pre-processing uncertainties for data assimilation and the RE on 2018-09-08 for qa > 0.75. (a) superobservation uncertainty constructed for this research (b) fully correlated uncertainty (c) representativity error (d) uncorrelated uncertainty.





reduction of 2.7%. Over polluted areas with a tropospheric $NO_2$ column over 30 $\mathrm{\mu mol\, m^{-2}}$ this reduction is 5%. With the

tiling approach, we avoid such a systematic low bias. The randomly sampled observations provide a noisy picture of the data, making it much less reliable than the other methods, demonstrating the large sub-grid variability.

The spatial structure of the superobservation uncertainty is illustrated in Fig. 13a, and is compared to two simplified methods of calculating the superobservation uncertainty. The associated RE is shown separately in Fig. 13c. Note how the RE is mainly present at the edges of cloud fields due to the low coverage there. Also, note how the RE is higher in high $NO_2$ areas due to the higher variation in measurement in these areas. This is particularly visible over Tunis and the Ruhr area.

The assumption that the observational uncertainty is fully correlated in space results in the uncertainties shown in Fig. 13b. Uncertainties using this approach are much higher than Fig. 13a and are likely overestimated. Assuming the uncertainty is fully uncorrelated results in a much lower uncertainty, as shown in Fig. 13d. In this case, the total uncertainty is dominated by the number of observations in the grid cell, somewhat reflecting the RE. This is a strong underestimation as compared to the uncertainty shown in panel 13a.

### 7.1 Data assimilation experiments

The impact of superobservations and their uncertainties on $NO_2$ analysis from $NO_x$ emission optimization is evaluated in a state-of-the-art chemical data assimilation framework. The data assimilation system used is described in Sekiya et al. (2022), which used the CHASER 4.0 chemical transport model (Sudo et al., 2002; Sekiya et al., 2018) at 1.125x1.125 degree resolution as the forecast model and the local ensemble transform Kalman filter (LETKF) data assimilation technique (Hunt et al., 2007). The detailed description of the assimilation approach used in this system is described in Miyazaki et al. (2020b).

To demonstrate the impact of different superobservation settings the following observational data for July 2019 were used in data assimilation sensitivity runs.

1. The superobservations and their uncertainties (figures 12a, 13a)

2. The superobservations with uncorrelated errors: The standard superobservations, with modified uncertainty assuming that the observations are fully uncorrelated in space (fig. 13d).

3. The superobservations with correlated errors: The standard superobservations with modified uncertainty assuming that the individual observations are fully correlated in space (fig. 13b).

4. Thinning: Thinned observations for which the values of one superobservation were taken randomly as one of the available observations within a model grid-cel, similar to figure 12d. The uncertainty is the corresponding retrieval uncertainty of this observation.

Note that the RE for thinned observations is expected to be higher than the standard superobservations. Nevertheless, the RE was set to be the same among experiments to assess the impacts of the superobservation uncertainty itself.

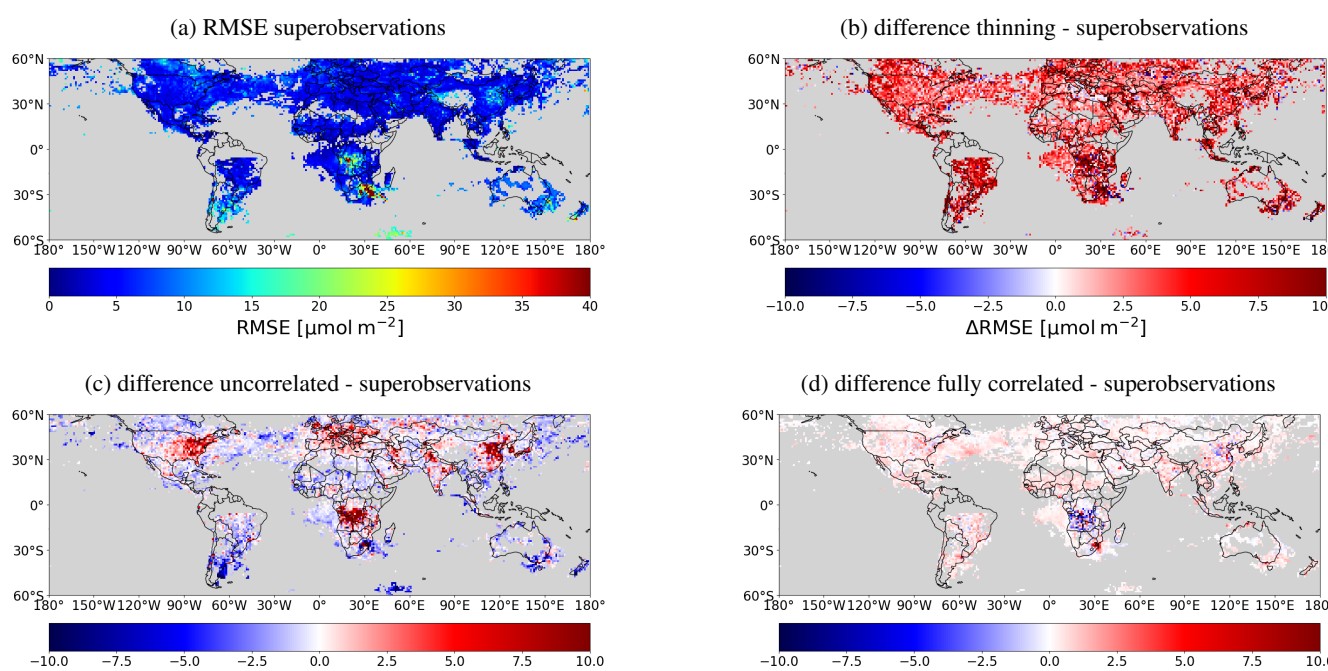

**Figure 14.** Panel showing the RMSE of the observation - forecast for the superobservations and how it compares to the other experiments. The RMSE is calculated over the time dimension, using only grid-cells for which the tropospheric column is over 17 µmol m$^{-2}$ (eq. 20 (a)) RMSE of the superobservations. (b) difference RMSE thinning - superobservations. (c) difference RMSE uncorrelated - superobservations (d) difference RMSE fully correlated - superobservations.

The effectiveness of the assimilation for these four experiments was evaluated with the observation-minus-forecast RMSE,

$$RMSE_{x,y} = \sqrt{\frac{1}{t}\sum_1^t (O_{t,x,y} - F_{t,x,y})^2} \qquad (20)$$

and the results are shown in fig. 14. The relative adjustments made by data assimilation is evaluated by comparing the analysis ($A_t$) and forecasts ($F_t$) as follows:

$$Impact[\%] = \frac{1}{t}\sum_1^t \frac{|A_t - F_t|}{F_t} * 100 \qquad (21)$$

The results are shown in fig 15. Additionally, the mean absolute difference (MAD),

$$MAD = \frac{1}{n}\sum_1^n \left| \frac{1}{t}\sum_1^t O_{t,x,y} - F_{t,x,y} \right| \qquad (22)$$

RMSE and $\chi^2$ metrics were evaluated for the different experiments, see table 1. The $\chi^2$ value is the ratio between the observation-forecast errors (actual errors) and the model plus observational uncertainties (estimated uncertainty). A $\chi^2$ of



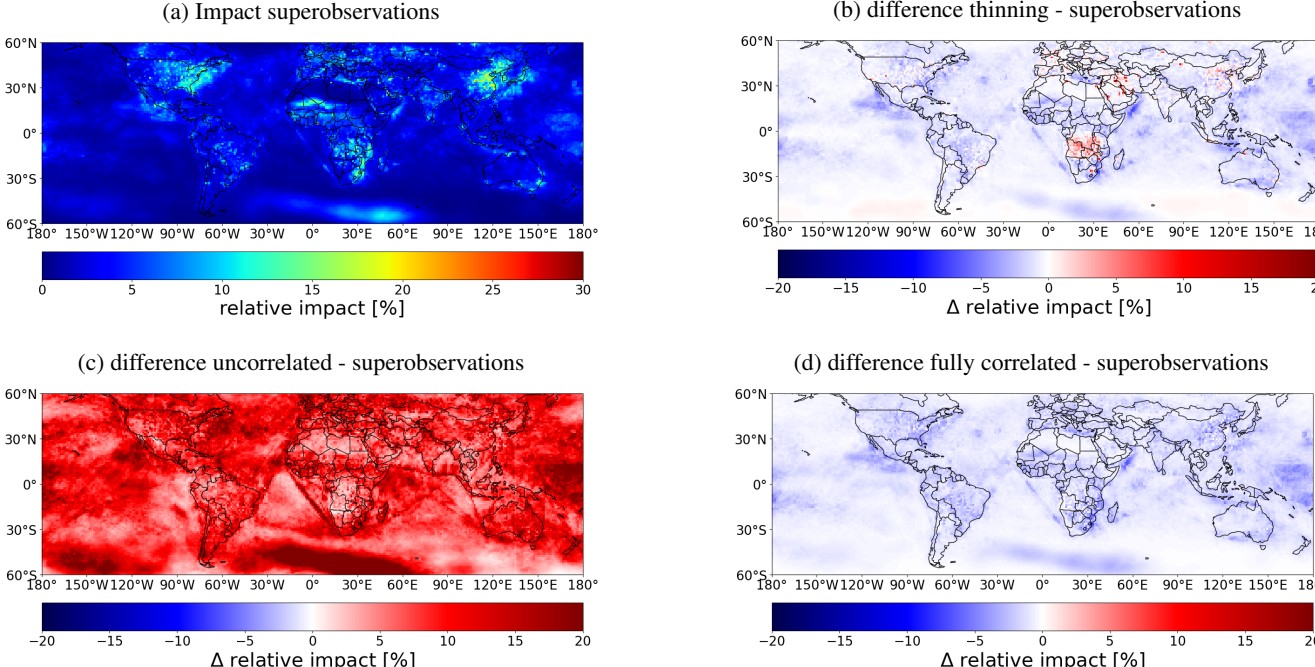

**Figure 15.** Panel showing the relative impact of the superobservations on the data assimilation system, and how this compares to the other experiments (a) relative impact of the superobservations. (b) difference relative impact thinning - superobservations. (c) uncorrelated - superobservations. (d) fully correlated - superobservations.

one means the residuals and uncertainties are balanced, while a higher chi-square value indicates uncertainties are underestimated and vice versa. It is calculated as in Sekiya et al. (2022); Zupanski and Zupanski (2006). $\chi^2$ was estimated only over

555 highly polluted areas with observation concentrations higher than 17 $\mu$mol m$^{-2}$, while the $\chi^2$ are somewhat different over regions with lower NO$_2$ concentrations. The impact calculation uses data between 11 and 17 hours local time, which is the time window when TROPOMI observations are available.

| Experiment | RMSE [$\mu$mol m$^{-2}$] | $\chi^2$[−] | MAD [$\mu$mol m$^{-2}$] |
|:---:|:---:|:---:|:---:|
| Superobs | 8.7 | 1.8 | 5.2 |
| Thinning | 15.7 | 1.7 | 9.5 |
| Uncorrelated | 12.1 | 111.0 | 4.8 |
| Correlated | 9.0 | 0.7 | 5.4 |

**Table 1.** Metrics of the data assimilation experiments



The $\chi^2$ value of the standard superobservations (Obs-1) is 1.8, which means either the model or observational uncertainties (or both) are somewhat underestimated. $\chi^2$ value can be sensitive to the choice of covariance inflation factor through its impacts on background error covariance (i.e. model errors), as indicated by Sekiya et al. (2022). We have conducted several sensitivity calculations by perturbing the covariance inflation factor and found that the impact on $\chi^2$ is limited because the increase in background error covariance is compensated by an increase in the observation-forecast error. The reason why $\chi^2$ is higher than 1 can be due to a variety of model forecast or observation errors that are not accounted for in the covariances, such as transport errors and error correlations between superobservations.

Both the MAD and RMSE are largest in the thinning case (Table 1). The decreased RMSE when using the superobservations indicates that averaging satellite observations leads to values which are closer to the scale that is represented by the model. The larger MAD in the thinning case reflects the fact that randomly selecting observations often results in negative tropospheric $NO_2$ columns, which are rejected by the assimilation system, resulting in a positive sampling bias. This effect is particularly obvious over remote areas with some negative values due to the retrieval uncertainties. In the case of superobservations, the proportion of observations with negative tropospheric columns and their value are both significantly reduced.

The standard superobservation case had the smallest RMSE compared to both the fully correlated and the uncorrelated cases. Given the common tropospheric $NO_2$ fields, the difference is attributed to the differences in the superobservation uncertainty. In the uncorrelated case, corresponding the smaller uncertainties, the data assimilation adjustments become larger than the standard superobservation case (fig. 15c), with larger RMSEs in highly polluted areas, probably due to overcorrections (Fig. 14c). In remote areas, the RMSE improves with smaller uncertainties, suggesting that the standard superobservations overestimate its uncertainty in remote areas. The smaller MAD in the uncorrelated case reflects the reduced RMSEs in remote areas.

On the other hand, in the correlated case, the uncertainty is large, which reduced the data assimilation impact and increased the RMSE. This shows that assuming the uncertainties are fully correlated is not so unrealistic, but it does lead to a reduction in performance. One exception to this is Central Africa, where the lower uncertainty improves the RMSE.

The uncertainty is similar between the thinning and fully correlated cases because the retrieval uncertainty is not as noisy as the retrieval concentrations. Correspondingly, the effect of the observations on the assimilation should be similar between the thinning and correlated cases. This is mostly true, except for some high-emission areas, such as central Africa, the Middle East and eastern China. Here the larger o-f and RMSE between the model and observations increases the impact of the observations on the assimilation. Also, note that thinning results in a similar $\chi^2$ value to the superobservations. The larger o-f and uncertainty maintain the ratio between the two.

## 8    Discussion

In this paper we presented a detailed methodology to construct superobservations and their errors and averaging kernels, improving upon the superobservations used previously by Miyazaki et al. (2012); Boersma et al. (2016); Sekiya et al. (2022); van der A et al. (2024). These superobservations are constructed in particular for data assimilation, inverse modelling and model evaluation applications. The first aspect of this is an improved estimation of the superobservation uncertainties stemming from



the observational uncertainties for individual TROPOMI $NO_2$ observations. This is achieved by quantifying the correlation between observations, allowing for a more accurate propagation of the observational uncertainties and the spatial distribution of these uncertainties. The spatial correlations for the slant column, stratosphere and air-mass factor contributions are estimated and treated separately. Uncertainties relating to the prior are not discussed because it is assumed the kernels will be used

during assimilation or model evaluation. As shown by the data assimilation experiments, realistic uncertainties are of key importance for the optimal performance of the assimilation system. The correlated experiment leads to an overestimation of the uncertainty. This is similar to the method of Inness et al. (2019b) and the HARP package spatial binning method for total uncertainty variables (http://stcorp.github.io/harp/doc/html/algorithms/regridding.html#spatial-binning). On the other hand, the uncorrelated experiment underestimates the uncertainty. It is similar to the HARP spatial binning method for random

uncertainty variables. Both an over- and underestimation of the uncertainty degrade the short-term forecast in the JAMSTEC data assimilation system, as demonstrated above.

The quantification of the spatial error correlation is complicated and remains uncertain. Correlations between retrieval uncertainties of nearby satellite pixels may be caused by spatially correlated biases in the characterisation of the surface reflectance or LER, aerosol and cloud properties and may depend on the weather. For instance, rainfall or drought may locally impact the

605 albedo, which is not described by the albedo climatology used in the retrieval. Estimating a correlation for the AMF uncertainty is particularly difficult because it results from complex interactions between algorithms and variables such as surface albedo, cloud albedo and cloud height, and unspecified systematic retrieval errors. The way these variables are spatially correlated propagates to the correlation of the AMF uncertainty.

The stratospheric uncertainty treatment was updated. For individual observations, the stratosphere does not contribute much

to the uncertainty, but for the clustered superobservations the stratosphere is a prime source of error. We quantified a longitude and seasonal dependent stratospheric uncertainty, replacing the default constant uncertainty present in the TROPOMI data product. As a result, lower latitudes have significantly lower stratospheric uncertainties. Uncertainties for the higher latitudes are generally lower than the default uncertainty but can also be higher depending on the season.

We also improved the existing method of calculating the (horizontal) RE. A simple constant parameterization was used be-

615 fore by Miyazaki et al. (2012); Boersma et al. (2016). We presented a mathematical derivation for the RE in the case of random missing observation. This allows for easier and more accurate computation of the RE. Additionally, we have quantified a systematic sampling correction for the case when the missing observations are clustered, as would be the case when clouds cover part of the superobservation area. This leads to higher uncertainties and a lower impact of low-coverage superobservations. The RE derivation also shows that a thinning approach (keeping just one observation per grid cell) would add a large uncertainty to

620 the observation equal to the standard deviation of the observations within a model grid cell.

Compared to Miyazaki et al. (2012), who postulated a fixed correlation of 0.15, our superobservations are somewhat more uncertain. However, due to the separation of the different components, the uncertainty correlation in our superobservations is spatially heterogeneous and has a different behaviour over the ocean than over polluted regions. In a further development (Sekiya et al., 2022) already separated the stratospheric error, treating it as fully correlated. However, this work still uses the

625 postulated correlation of 0.15 for the remaining observational uncertainties. This means that the slant column uncertainties





presented here are lower than theirs, but our AMF uncertainty is higher (except for very large superobservations). Compared to Sekiya et al. (2022) our superobservations are somewhat more impactful over clean areas, and somewhat less over polluted areas.

When compared to thinning the superobservations are a much less noisy representation of the satellite data, and thereby improve the performance of the data assimilation. The uncertainty-weighted superobservations also provide a realistic average of the data, but they favour the small column retrievals and are therefore low-biased, which is a feature we avoid using the tiling approach.

The superobservations resolve the correlations between observations within the superobservation grid cell. However, it does not describe a remaining correlation between adjacent superobservations. Inflating the superobservation uncertainty could improve the results of the assimilation depending on the size of the superobservations.

We have focused on constructing superobservations of the same size as the grid cell of the model they will be compared against. However, it is not obvious that this would be the most optimal configuration. According to Nyquist, in order to capture all the variability at the size of the superobservations we would need to oversample by introducing extra superobservations shifted in space. One may argue that for a species like $NO_2$ with a very inhomogeneous fine-scale distribution, interpolation in model space is not useful without knowledge of the subgrid distribution of the emission sources. Data assimilation systems typically introduce spatial correlation lengths covering multiple grid cells. They behave like low-pass filters and fine-scale features are not constrained in the analysis. In that case, making superobservations larger than a single grid cell, reflecting horizontal correlation lengths of the assimilation system could allow for a better representation of the large-scale structures which are constrained in the analysis and at the same time lowering computational costs.

## 9 Conclusion

In conclusion, this research has improved and formalized existing methods of creating superobservations. Superobservation uncertainties have been quantified by analysing the various aspects leading to systematic and random uncertainties in the satellite retrieval, and by mathematically deriving a realistic representativity error. Data assimilation experiments show that the uncertainties derived in this way lead to better forecast results than postulating either fully correlated or uncorrelated uncertainties. A thinning of the observations results in very noisy patterns of $NO_2$ and degraded assimilation results compared to the superobservations. Thus we recommend the use of superobservations with quantitative uncertainties for the assimilation of atmospheric $NO_2$ and other trace gases.

The superobservation methodology is generic, and will be applied in the future to other species, like HCHO, $SO_2$, CO, $O_3$, $CH_4$ and $CO_2$ and to other satellite instruments like OMI, GEMS or TEMPO. All of the concepts and mathematics described in this paper are broadly applicable. This includes the method of clustering, averaging and uncertainty averaging. The latter does require the quantification of correlations. Calculating the RE is also species-agnostic, with only the systematic correction requiring extra quantification.





Another possible application for superobservations is the creation of level-3 data. These methods provide satellite information on regular grids. Our superobservation approach provides realistic error estimates of the grid box mean value in case the
660 level-3 gridboxes contain multiple satellite footprints, and have a lower resolution than the satellite. This is a first step towards a consistent averaging of the satellite data into monthly, seasonal and yearly averages and specifying meaningful uncertainties for such averages. Additional considerations are needed to quantify the temporal representativity and temporal correlations. Also, in our work, targeting model comparisons and data assimilation, we did not consider the a-priori uncertainties which may need to be quantified for level-3 data, depending on the application.

*Code and data availability.* The code used in this paper to generate the superobservation data is available on Zenodo doi.org/10.5281/zenodo.10726644. The TROPOMI $NO_2$ L2 datasets used in this paper are made available operationally through the ESA Sentinel-5P data hub (https://s5phub.copernicus.eu, last access: 25 Jan 2023). The S5P-PAL NO2 dataset is available from the S5P-PAL website (https://data-portal.s5p-pal.com/products/no2.html, last access: 25 January 2023), and was generated with version 2.3 of the TROPOMI processor. Note that this dataset will be replaced by the latest reprocessing based on processor version 2.4.





## Appendix A: Correlation calculation

### A1 Calculation of the gridbox-mean correlation from the correlation length

The mean correlation $C$ between pairs of observations within a superobservation is calculated as an average of the correlation between all pairs of points in a superobservation. This is obtained by multiplying the probability density function (PDF) of the distance between all points in a square with the correlation as a function of the distance and integrating the result. The PDF of all points in a rectangle is taken from Philip (1991) and the correlations are calculated for a distance $d$ and a correlation length $l$ assuming an exponential decay, $C = e-d/l$. The correlation length $l$ is computed using the TROPOMI v2.4 - v2.3.1 differences.

For example, a 1-degree superobservation at 29 degrees latitude is a 113 by 99 kilometre rectangle resulting in a PDF as shown in Fig. A1a. A correlation length of 32 kilometres yields Fig. A1b. Multiplying the two functions creates figure A1c, and integrating this results in a correlation of 0.24 (Fig. A1d).

### A2 Calculation of the correlation length

The correlation length is calculated using the inverse of the method for calculating the correlation, where a correlation is converted to a correlation length based on the PDF of distances. This requires a representative correlation of the dataset, with a representative grid-cell. The autocorrelation of the v2.4 - v2.3.1 tropospheric column differences within a 1-degree superobservation is calculated using equation A1. This is a special case of the Pearson correlation where we assume the mean of the v2.4-v.2.3.1 retrieval difference is 0.

$$C_{x,x} = \frac{\sum_i^n \sum_j^n x_i x_j}{\sum_i^n x_i^2} \tag{A1}$$

Correlations are computed within single superobservations. Figure A2b shows the difference and the correlation on a single day. Averaging the correlation over the available data gives figure A2c. Because the AMF is most important for polluted observations we filter this result for areas with an average tropospheric $NO_2$ concentration over $30\,\mu mol\,m^{-2}$, figure A2d. The average correlation of the remaining data is 0.244 at an average grid-box size of 113x99 $km^2$. Applying the inverse method of section A1 gives a correlation length of 32 kilometres.

## Appendix B: Modified representation error

The RE equation B1 is modified for better usage with the effective population size ($N_{eff}$). The fitting of an effective number of observations results in equation B1, where $n$ is replaced by an effective number of observations $n_{eff}$. Note that the correction for sampling without replacement is not modified. The number of effective observations $n_{eff}$ is calculated using the total number of effective observations $N_{eff}$ and the coverage $f$ (eq. B2), which yields equation B3.







**Figure A1.** Panels showing the computation of the superobservation correlation from a correlation length and latitude. (a) The probability density function for the distance between two points in a 1-degree superobservation at 29 degrees latitude. (b) Correlation function for a correlation length of 32 kilometres. (c) Multiplication of the PDF in panel a and the correlation function in panel b. (d) Integration of the curve from panel c. The area under the curve gives the correlation of the superobservation.





(a) Difference in troposhperic $NO_2$ v2.4-v2.3 ,2018-09-14.

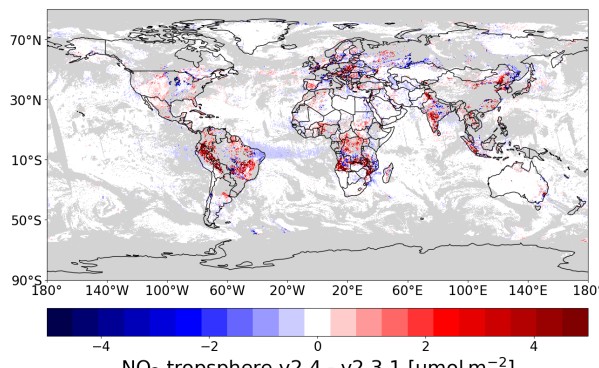

(b) Correlation 1-degree superobserations 2018-09-14.

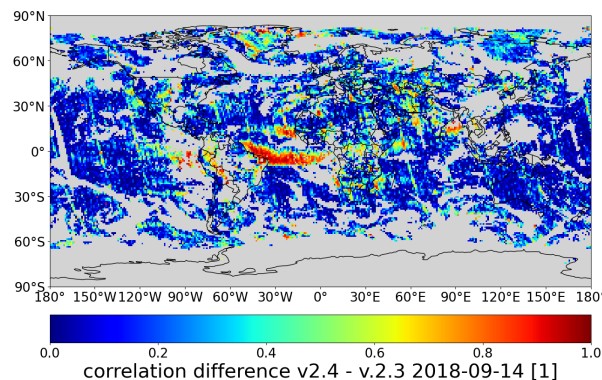

(c) Mean correlation 1-degree superobserations.

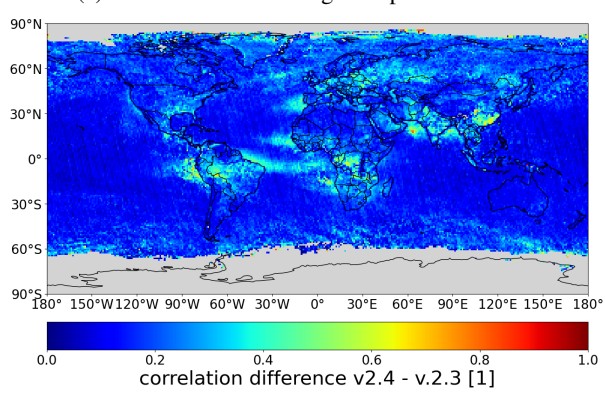

(d) Polluted areas mean corelation.

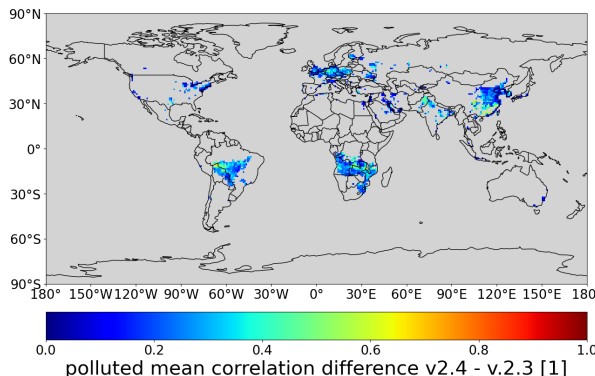

**Figure A2.** Panels showing the calculation of a representative correlation for the purpose of calculating a representative correlation length. (a) difference between the v2.4 and v2.3 observations. (b) Correlations within 1-degree superobservations relative to dataset. (c) Average correlation in the month of September for 2018-2022. (d) Average correlation of polluted areas.

$$\sigma_{RE,n} = \frac{\sigma}{\sqrt{n_{eff}}} \sqrt{\frac{N-n}{N-1}} \tag{B1}$$

$$n_{eff} = N_{eff}f \tag{B2}$$

$$\sigma_{RE,n} = \frac{\sigma}{\sqrt{N_{eff}f}} \sqrt{\frac{N-n}{N-1}} \tag{B3}$$





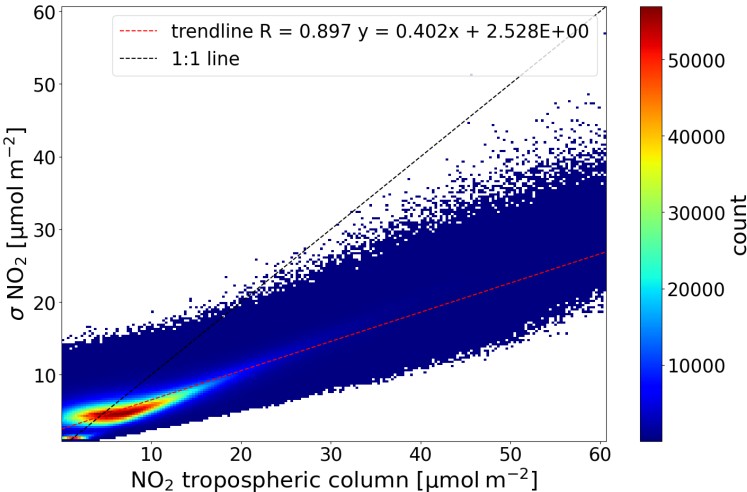

**Figure C1.** Relationship between the superobservation tropospheric column and standard deviation for a 0.3 degree superobservation.

However, Because $N_{eff}$ is smaller than N, applying equation B3 results in a representativity error larger than $1\sigma$ for lower coverages. This is not consistent with the experimental data shown in figure 8c and as a result, the fit of the number of effective observations is not optimal. Adding a plus one to the equation solves this, resulting in equation B4.

$$\sigma_{RE,n} = \frac{\sigma}{\sqrt{N_{eff}f+1}}\sqrt{\frac{N-n}{N-1}} \tag{B4}$$

If we assume $N$ is large the equation can be further simplified to equation B5.

$$\sigma_{RE,n} = \frac{\sigma}{\sqrt{N_{eff}f+1}}\sqrt{1-f} \tag{B5}$$

While this equation is not necessary for fitting a single superobservation, it makes the equation independent from the original number of observations in the superobservation. This simplifies the averaging of the $\sigma$ normalized RE of a superobservation over multiple orbits and fitting the number of effective observations of that superobservation location.

**Appendix C: Fallback standard deviation**

For the case when there are insufficient observations to calculate a meaningful standard deviation for a grid cell we implemented a fallback option where the superobservation standard deviation is estimated as 0.4 times the tropospheric column + 3 $\mu\mathrm{mol}\,\mathrm{m}^{-2}$, based on the trendline in figure C1. This relationship has a Pearson correlation of 0.9. We calculated the fallback using 0.3 degree superobservations, which contain on average 25 TROPOMI observations at the equator.



*Author contributions.* HE and KFB wrote the original Fortran software package to generate superobservations. AD and PR further developed this code, and PR developed a Python version. PR and HE wrote most of the text of the paper, and PR made most of the images. TS and KM helped with the setup of the assimilation system and the interpretation of the results from the system. SH contributed ideas and discussions for estimating the observational correlations and the RE.

*Competing interests.* The contact author has declared that none of the authors has any competing interests.

*Acknowledgements.* Sentinel-5 Precursor is a European Space Agency (ESA) mission on behalf of the European Commission (EC). The TROPOMI payload is a joint development by ESA and the Netherlands Space Office (NSO). The Sentinel-5 Precursor ground segment development has been funded by ESA and with national contributions from the Netherlands, Germany, and Belgium. This work contains modified Copernicus Sentinel-5P TROPOMI data (2018–2023), processed in the operational framework or locally at KNMI.

HE acknowledges financial support from the CAMEO project (grant agreement No 101082125) funded by the European Union. Views and opinions expressed are however those of the author(s) only and do not necessarily reflect those of the European Union or the Commission. Neither the European Union nor the granting authority can be held responsible for them.



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
