# Peer review of "Quantifying uncertainties of satellite $NO_2$ superobservations for data assimilation and model evaluation"

_EGUsphere, 2024_

## Referee Comment (RC2)

**Summary**

The goal of this study is to enhance the supraobservation technique, which aggregate satellite remote sensing $NO_2$ observations, to better align with the model resolution. The authors carefully quantify and formulate the observation error budget, which includes the error caused by the total slant column fitting, the stratospheric column, the AMF calculation, and the spatial representativeness. The author also conducted several data assimilation experiments to examine the benefits of using superobservation, concluding that the proposed superobservation method can lead to the lowest forecast error compared to other approaches.

As a result, the superobservation approach presented in this study is important to the research community and fits within the scope of GMD, and this algorithm has the potential to be extremely useful for preprocessing high spatiotemporal resolution trace gas observations from geostationary instrument. Overall, though the topic is important, there are several issues that need to be addressed before considering its publication.

**General comments**

Many abbreviations in the manuscript are not clearly defined (e.g., TROPOMI, TEMPO, GEMS, LER/DLER, GFED, JAMSTEC, etc.). I would recommend authors to double-check and make adjustments to the manuscript.

Line 205-208: The statements presented here are statistically correct, and the authors do not need to change them. I just want to suggest another way to think about the negative $NO_2$ column. As the author noted in the manuscript, the appearance of a negative $NO_2$ column is primarily due to limitations in the retrieval algorithm, which could be improved in the future. The biggest issue, in my opinion, is that negative $NO_2$ columns have no physical meaning, and they should actually correspond to small positive $NO_2$ columns in non-polluted areas. Taking this into account (i.e., non-negative trace gas concentration), a "negative bias" may occur when averaging negative and positive pixels.

Section 6.2: It is quite impressive to see the author put in so much effort to quantify the representative error that result from data coverage in a superobservation grid, and I really think this section is the gist of this study. However, the writing in Section 6.2 is disorganized and difficult to follow. I would encourage the authors to enhance the writing in this section, particularly the explanation/derivation of effective population size.

The level 2 product also reports $NO_2$ tropospheric column uncertainty

(nitrogendioxide_tropospheric_column_precision). Does this term contribute to the error calculation in this study? I didn't see any discussion about it; am I missing something?

**Specific comments**

Line 195-197: Given the same observation uncertainty, the Kalman gain only depends on background error. As a result, low and high $NO_2$ observations should have the same Kalman gain, contradicting your claim that low $NO_2$ observations force more in the assimilation than high $NO_2$ observations and yield low-biased results. Could authors elaborate on this?

Figure 3: This figure is provided in the manuscript, but it is not mentioned in the content.

Equation 13: Please define $\Theta$ and $\Theta_0$.

Line 309: Please define GFED and provide a reference for this fire emissions inventory.

Line 360-362: Is there other dataset that might be utilized to estimate spatial correlation lengths? I ask this because the version difference may not always be available.

Line 372: Could you clarify which variable you referred to as "uncertainty due to the AMF, as estimated by the retrieval" in the level 2 product?

Section 5.3: As albedo has seasonal variation, I am wondering if the AMF error spatial correlation length also has a significant seasonal change.

Section 5.3: What is eventually used for the AMF uncertainty is not clear. Is the uncertainty coming from the version difference or from the retrieval product?

Line 396-397: Is the green line just the average of the gray lines in Figure 8?

Line 421-423: It would be beneficial to further explain the calculation of fractional coverage ($f$), for example, by utilizing equations.

Line 438-439: "At 50% coverage, the increase in RE is 54% for clean areas and 263% for polluted areas." Figure 8 does not support this statement. This sentence likely describes the inflation of representative error due to partially data coverage in Figure 9. Please consider moving this sentence to the right place.

Equation 18: This equation is provided but doesn't mention in the manuscript.

Line 459: Could the author clarify what "sigma normalized RE" means here?

Line 485-490: I am confused with these sentences. Did you mean that you first estimated $\sigma$ using Equation 16 and then computed the final representative error using Equation B5? Could author clarify on it?

Section 6: Many regional chemical data assimilation (DA) systems employ a finer horizontal grid (~ 10 km or less), which is comparable to the scale of a satellite pixel. In this scenario, the proposed method for estimating representative error may not be effective because the pixel population ($N$) becomes too small, leading to the use of the climatology value instead. I was wondering if the author could comment on the extent to which we should consider the representative error.

Line 494-495: Where did the intersection come from? I believe it is from Figure C1. Is that correct?

Line 497: "uncertainty$^2$" Please fix the typo.

Line 509: Please correct typo in this sentence for the figure citation.

Line 515-516: It is acceptable to retain all the descriptions here. I just want to comment on the fact that the random sample approach is not a very good method for data thinning given the large superobservation grid. A better way is to first analyze the distribution of all pixels inside a superobservation grid and then select one pixel that is closest to the mean or median. This method could pick up a more statistically representative data than a random sampling.

Section 7.1: In light of the previous comment, it is not surprising that the thinning experiment demonstrated the worst performance in DA. A data-thinning experiment is too easy to outperform as a large discrepancy between model and superobservation is expected. Keeping this experiment in the manuscript is totally fine. I would like to encourage the authors to run one more experiment in which the superobservation (error) is just a simple average of all pixels' $NO_2$ columns (reported error) that falls within the superobservation grid. Given that this approach is commonly used in many DA or data analysis studies, comparing the method proposed in this work to the "simple average approach" could help to highlight the benefit of the using advanced superobservation method. I recognize that it may be difficult for the author

to run additional simulation, thus this is not required. Alternatively, it would be wonderful to see some discussion of the simple average method versus a more complicated superobservation method in the manuscript.

Table 1: Why is the $\chi^2$ value for the uncorrelated error case (111.0) much larger than in other experiments? Could author verify this?

Table 1: The RMSE for fully correlated error and superobservation cases is similar. The $\chi^2$ value for the fully correlated error case is actually closer to one, which makes it difficult to conclusively say that the superobservation case outperform the fully correlated error experiment.

Line 583-584, Please change the "o-f" to "O-F".

Section 7.1: The error budget analysis (Figure 11) shows that representative error is not the dominant source of the observation uncertainty. As the computation of representative error seems to be very costly, did the author have a chance to examine the impact of including or excluding representative error on the DA results?

Section 7.1: I appreciate that the author conducted a thorough comparison between different DA experiments. Since the RMSE of many DA experiments are comparable, I would recommend finding an independent dataset for model evaluation to better understand the performance of individual experiments and determine which experiments better capture the real $NO_2$ fields. An observing system simulation experiment (OSSE) might also be useful in future work to test the performance of different superobservation techniques, since the ground truth is known.

Line 600: Please define JAMES.

Line 625: I am confusing about using correlation 0.15 for remaining observation error. This contradicts what is stated in Section 5.2, where the correlation is determined as a function of distance/correlation length, and each grid may have a different value for the AMF uncertainty. Could the author please clarify this?

Line 637: "According to Nyuist". It would be recommended to add a reference here.

Line 640-644: The spatial correlation imposed on prior emissions and concentrations may be incorrect, and they are also updated by the DA process. As a result, using the predefined spatial

correlation to justify the use of superobservations is not a good claim in my opinion.

Line 676: Please correct the typo in the equation.

Equation A1: Please define $x, i, j$, and $n$ in the equation.

Line 712: The intercept in the figure is +2.5, not +3 mol m$^{-2}$ written in the manuscript.

---

## Author Comment (AC1)

**Reply RC1 (DOI: 10.5194/egusphere-2024-632)**

**Summary**
This study investigated the superobservation methodology for satellite observations, especially for chemical tracers. The paper discussed how to construct superobservations and appropriately set their uncertainty. The authors took several aspects into account and visualized their contributions to the resulting superobservations. They also performed data assimilation experiments and showed that their superobservation methodology resulted in improved forecasts compared to simple thinning.

The paper appears to align with the scope of GMD and may have significant implications for data assimilation studies involving satellite observations. Unfortunately, certain points were not clear to me and could benefit from clarification prior to publication. In addition, the paper's readability was somewhat challenging, possibly due to its unconventional structure. Therefore, I would like to recommend Major revisions.

> Dear reviewer, we would like to thank you for your time, effort and well substantiated comments. Below our reply to your comments, we have tried to incorporate as many of your suggestions as possible. We are in particular committed to improve the readability of the paper. All line numbers in the reply refer to the revised manuscript.

**Major comments:**

**1. Clarifications required**
Several statements in the manuscript were unclear to me. Most of them could be my misunderstanding, but I would like to ask the authors for enhancing the clarity.

Lines 196–197: The sentence beginning with "Given the same uncertainty," is confusing. Could this be re-written?

> Due to changes in the structure of the paper this point has been moved to the introduction on line 85-88. The sentence has been rewritten to:
>
> "If all individual observations with their individual uncertainties are assimilated in a model with a coarser resolution than the satellite, this leads to low-biased analyses, because more weight is given to low observations with a small uncertainty. With the superobservation approach described in this paper, such persistent low biases are largely avoided."

Lines 322–323: Unfortunately, I could not understand this sentence. A rewrite may be necessary.

This sentence and the previous sentence (lines 320-321) have been rewritten to:

"Any systematic error on the slant column also influences the quantification of the stratospheric error discussed in the previous section because the slant column is assimilated for the quantification of the stratosphere."

Lines 510–511: This sentence was confusing. I guess this sentence compares Figures 12a and 12c, yet I could not find clear differences between these figures.

The difference between the figures is subtle, but systematic. To make this more clear we rewrite lines 493-499 to:

"The regular superobservations and the uncertainty superobservations are similar. Both give a realistic low-resolution representation of the original satellite data. But, as expected, the uncertainty-weighted superobservations have systematically lower values because the weights favour the smaller columns, though the difference remains subtle. This is most clearly observed above Paris and North Africa. On average the uncertainty weighted superobservation in Figure 12 have a tropospheric column of 22.4 μmol m$^{-2}$, compared to 23.0 μmol m$^{-2}$ of the normal superobservations, which is a reduction of 2.7%. Over polluted areas with a tropospheric NO2 column over 30 μmol m$^{-2}$ this reduction is 5%."

Lines 561–562: Why does an increase of covariance inflation result in a larger O–B?

Inflating the covariances increases the spread in model results, which in turn impact the analyses. The analyses with an increased covariance inflation produced a lower quality forecast. This can be expected because a too large ensemble spread can degrade the comparison against observations. On the other hand, if the spread is too small, an increase in the inflation may lead to a reduction of the departure.

To clarify this, we add the following changes to the text on line 553-554:

"The increase in spread from the covariance inflation results in a poorer forecast."

Line 641: What does correlation length mean? The cutoff radius of localization?

In this case the correlation length refers to the spatial correlation lengths introduced in the modelling of the background (forecast) coveriance matrix **B**.

To clarify this, we replace the sentence on lines 640-642 by:

> "Data assimilation implementations typically introduce spatial correlation lengths covering multiple grid cells in the modelling of the background (forecast) covariance matrix **B**."

Line 703: I am not sure why adding one improves the results. I understood that N > Neff but does this sentence mean N–Neff=1? Could you explain? Furthermore, I could not understand that "This is not consistent with the experimental data" in Fig. 8c.

> Because $N_{eff} < N$, there will always be a point where $N_{eff} * f < 1$. Normally this does not happen because you have at least one observation n. And for n=1 the systematic solution should be equal to the random solution because the sampling method is still the same. Yet for the coverage where $N*f = 1$, $N_{eff} * f < 1$, this yields an n lower than 1. Resulting in $\sigma_{re} < 1\sigma$. Experimentally this never happens and conceptually this should also not happen. Adding the plus one was a simple way to address this problem, while keeping a continuous function. Based on other comments we have decided to rewrite the section on the representation error. In this rewrite we have more formally derived the need for adding a plus one, which can be found in appendix B. This derivation results in a slightly different formula. Note that in the new derivation the plus one is already added in the random representation error formulation to address problems with the use of fractional observations. This solves the above problem before it occurs

**2. The structure of the paper**

The manuscript does not have a typical structure that contains an introduction, method, result, discussion, and summary. Although the motivation of the study should be clearly noted in the first section, section 3 explains the motivation of superobservation as well. While section 5 discussed contributions of three aspects on uncertainty, section 6 revisits the topic of uncertainty. I might misunderstand, but I would like to ask the authors to re-consider the structure of the manuscript. This could make the paper more concise.

> Because of the nature of the study where the method is our result, we found a more typical structure did not fit the contents, which is why we decided on using an unorthodox structure. But not following a traditional structure does carry additional risk in terms of understandability, thus we appreciate the feedback on this topic. To improve the readability, we have merged section 3 into the introduction. Point 1 can now be found on line 44, point 2 on line 74, point 3 on line 84 and point 4 on line 83, point 5 on line 83 and point 6 on line 89

> While both section 5 and 6 discuss the superobservation uncertainty, the sources of the uncertainty are different. Section 5 addresses the dependence of the superobservation uncertainties on the uncertainties of the individual satellite retrievals and spatial correlations between these errors. Section 6 discusses the representation error which is not linked to uncertainties in the individual retrievals but to an incomplete sampling of the grid box. These require a fundamentally different treatment and keeping these separate improves the readability of the paper.

**Minor comments**

Line 92: Remove an extra "observational?"

 We have removed the extra observational

Line 238: Does the left-hand side correspond to ds?

 This is correct, we have added this to formula 8.

Line 246: Does $A_S$ mean the superkernel?

 Yes, for clarity we have added As to line 242.

Line 299: What are $\Theta$ and $\Theta_0$?

 These are the viewing zenith angle (VZA) and the solar zenith angle (SZA) respectively. This has been added to the text in line 289 for clarity.

Line 531: I would recommend explaining the experimental settings a bit more. It would be better to include what observations were assimilated, how long the assimilation window, how localizations were set, and how covariance inflation was achieved.

 We add the following information on the experimental setting to the text on line 516:

 "The assimilation was performed with 32 ensemble members and a two-hour assimilation window. Covariance localization was applied based on species-dependent localization scales, that were derived from sensitive tests in Miyazaki et al. (2012b). Covariance inflation was also applied by inflating emission factor uncertainties(ie. ensemble spread), to a minimum predefined value. Additionally, a multiplicative covariance inflation of 7% was applied to the concentrations. In addition to NO2, the assimilated measurements included total columns from the thermal-infrared (TIR)/near-infrared (NIR) band of the Measurement of Pollution in the Troposphere instrument (MOPITT) (Deeter et al., 2017), OMI $SO_2$ planetary boundary layer vertical columns(Li et al., 2020), and Aura Microwave Limb Sounder (MLS) $O_3$ and $HNO_3$ profiles(Livesey et al., 2022). To demonstrate the impact of different superobservation settings the following 4 sensitivity runs were done for July 2019, only varying the $NO_2$ observations:"

Line 551: The $X^2$ metrics seem similar with the consistency ratio (Dowell and Wicker 2009, 10.1175/2008JTECHA1156.1). Are they the same?

 The consistency ratio is very similar to the $X^2$ metric. Both serve the same purpose of assessing the balance between uncertainties and innovations. One crucial difference is that for $X^2$ the innovation ends up in the numerator while for the consistency ratio it is part of the denominator. As a result, they are like each other's inverse.

---

## Author Comment (AC2)

**Reply RC2:**

**Summary**

The goal of this study is to enhance the superobservation technique, which aggregate satelliteremote sensing NO2 observations, to better align with the model resolution. The authors carefully quantify and formulate the observation error budget, which includes the error caused by the total slant column fitting, the stratospheric column, the AMF calculation, and the spatial representativeness. The author also conducted several data assimilation experiments to examine the benefits of using superobservation, concluding that the proposed superobservation method can lead to the lowest forecast error compared to other approaches.

As a result, the superobservation approach presented in this study is important to the research community and fits within the scope of GMD, and this algorithm has the potential to be extremely useful for preprocessing high spatiotemporal resolution trace gas observations from geostationary instrument. Overall, though the topic is important, there are several issues that need to be addressed before considering its publication.

> We thank the reviewer for the positive feedback on the importance of our study, and well substantiated comments which are helpful to improve the paper. Below our replies to your comments. All line numbers refer to the revised manuscript.

**General comments**

Many abbreviations in the manuscript are not clearly defined (e.g., TROPOMI, TEMPO, GEMS, LER/DLER, GFED, JAMSTEC, etc.). I would recommend authors to double-check and make adjustments to the manuscript.

> We have added definitions of all abbreviations throughout text.

Line 205-208: The statements presented here are statistically correct, and the authors do not need to change them. I just want to suggest another way to think about the negative NO2 column. As the author noted in the manuscript, the appearance of a negative NO2 column is primarily due to limitations in the retrieval algorithm, which could be improved in the future. The biggest issue, in my opinion, is that negative NO2 columns have no physical meaning, and they should actually correspond to small positive NO2 columns in non-polluted areas. Taking this into account (i.e., non-negative trace gas concentration), a "negative bias" may occur when averaging negative and positive pixels.

> We agree with the referee that negative concentrations are not physical. But in a statistical sense they occur naturally. $NO_2$ concentrations cover a wide range of values, going from the polluted boundary layer to the stratosphere and remote regions like the Southern Hemisphere Pacific Ocean. The retrieval of the tropospheric column involves subtracting two large numbers, the total column coming from the measurement, and an estimate of the stratospheric background from the TM5-MP model and data assimilation system. Because of the subtraction of two large numbers with small relative errors, a Gaussian spread of errors is realistic and for instance a log-normal distribution of positive values would not make sense. The probability density function (PDF) of possible tropospheric column values will have a spread determined by the errors of both quantities. When the mean is much smaller than the

spread or uncertainty, we expect a large number of negative values to occur (over clean regions like remote ocean areas), while the mean of the PDF is still above (and close to) zero. Removing all the negative values will lead to a systematic positive bias of the order of the retrieval uncertainty over remote regions, which we want to avoid and justifies our approach.

We show that the construction of superobservations averages the positive and negative values, reducing the relative percentage of negative observations. This is one advantage of the superobservation approach.

Chemical solvers in models cannot cope with negative concentrations, and clipping will be needed in data assimilation applications.

Section 6.2: It is quite impressive to see the author put in so much effort to quantify the representative error that result from data coverage in a superobservation grid, and I really think this section is the gist of this study. However, the writing in Section 6.2 is disorganized and difficult to follow. I would encourage the authors to enhance the writing in this section, particularly the explanation/derivation of effective population size.

Based on your feedback and the feedback from the other reviewer we have decided to do a major rewrite of section 6 (now section 5), in which we in particular keep in mind readability and the derivation of the effective population size.

The level 2 product also reports NO2 tropospheric column uncertainty (nitrogendioxide_tropospheric_column_precision). Does this term contribute to the error calculation in this study? I didn't see any discussion about it; am I missing something?

The total precision reported in the level 2 product is comprised of the three different uncertainty components discussed separately in this paper (for more information see the TROPOMI $NO_2$ ATBD). As explained, we have replaced the stratospheric column precision by a more detailed analysis because of the importance for the superobservation error, but apart from this the combined precision is consistent with the "nitrogendioxide_tropospheric_column_precision". We aim to make this more clear by rewriting line 264 to 269:

"As shown in equation 2 the tropospheric column uncertainty consists of 3 separate sources of uncertainty: The stratospheric uncertainty, the slant column uncertainty and the air mass factor uncertainty. The superobservation uncertainty of these components is calculated separately because they will have different spatial error correlations, which means their uncertainty propagates differently. Every component is discussed individually in the sections below. Note that these components are not provided separately in the retrieval data file, but using the methods from the algorithm theoretical basis document (ATBD) (Geffen et al. 2022b) they can be reconstructed using the available information."

**Specific comments**

Line 195-197: Given the same observation uncertainty, the Kalman gain only depends on background error. As a result, low and high NO2 observations should have the same Kalman gain, contradicting your claim athat low NO2 observations force more in the assimilation than high NO2 observations and yield low-biased results. Could authors elaborate on this?

> You are correct that given the same uncertainties the low and high NO2 observations will force the same. The original text was confusing and has been revised. Because high NO2 observations have a higher uncertainty they will force less than low NO2 observations. This results in low-biased analyses. Averaging the observation into superobservations reduces this bias significantly.

> Based on comments from the other reviewer the information of this section has been reformulated. Note that the text from section 3 has been moved to the introduction (in response to reviewer 1), and this point can now be found in line 84:

> "The uncertainty of individual observations often scales with the column amount. This is the case for NO2 column retrievals, related to uncertainties in the air-mass factor. If all individual observations with their individual uncertainties are assimilated in a model with a coarser resolution than the satellite, this leads to low-biased analyses, because more weight is given to low observations with a small uncertainty. With the superobservation approach described in this paper, such persistent low biases are largely avoided."

Figure 3: This figure is provided in the manuscript, but it is not mentioned in the content.

> We have added a reference to the figure on line 218-220:

> " The weights $w_i$ are obtained by covering (tiling) the grid box with the TROPOMI observations, as shown in Fig. 2. They are equal to the area overlap between the footprint of the TROPOMI observation $y_i$ and the selected model grid box. An example of this method is shown in figure 3."

Equation 13: Please define $\Theta$ and $\Theta$ 0.

> These are the viewing zenith angle (VZA) and the solar zenith angle (SZA) respectively. This has been added to the text in line 287-290 for clarity:

> "To improve on this, we analyse the observation – forecast (OmF) departure between the TROPOMI and model column, using a geometric air-mass factor for both (eq. 13) using solar zenith angle $\Theta_0$ and viewing zenith angle $\Theta$)."

Line 309: Please define GFED and provide a reference for this fire emissions inventory.

> We have added a definition and reference to GFED one line 308-309:

> "In TM5-MP these are based on climatological fire intensities from the Global Fire Emissions Database (GFED)(van der Werf et al., 2017),"

Line 360-362: Is there other dataset that might be utilized to estimate spatial correlation lengths? I ask this because the version difference may not always be available.

We sympathise with the concerns of the referee concerning reproducibility. ESA and EU-Copernicus have decided to remove older retrieval data versions and test datasets once newly reprocessed data has become available. The DDS test datasets can be made available by the KNMI on request. For experienced researchers it is possible to recalculate the AMF using the different surface albedos (which are available publicly), which would effectively reproduce our results.

Line 372: Could you clarify which variable you referred to as "uncertainty due to the AMF as estimated by the retrieval" in the level 2 product?

The uncertainty in the level 2 product is not separated into its components. We have followed the information provided in the ATBD on how the uncertainty is computed. This means that the uncertainty due to the AMF must be calculated from the data available in the files. We clarify this on line 264-268

"The superobservation uncertainty of these components is calculated separately because they have different correlations, which means their uncertainty propagates differently. Every component is discussed individually in the sections below. Note that these components are not provided separately in the retrieval, but using the methods from the algorithm theoretical basis document (ATBD) (van Geffen et al.,2022a) they can be reconstructed using the available information."

Section 5.3: As albedo has seasonal variation, I am wondering if the AMF error spatial correlation length also has a significant seasonal change.

The seasonal dependence of the correlation length is neglected. The dataset we were able to use only contained data for September, which is the main reason why we did not quantify the seasonal dependence. It is likely there is some seasonal component to the AMF error correlation length. Because the correlation length is relatively uncertain it is a global average, which means seasonal effects are somewhat dampened. Thus, we believe the impact of not quantifying the seasonal variation in the correlation length is small. Still this would ideally be quantified in future work. In that case the latitudinal variations should also be quantified, as latitude has a major impact on seasonal variations.

Section 5.3: What is eventually used for the AMF uncertainty is not clear. Is the uncertainty coming from the version difference or from the retrieval product?

We use the uncertainty from the retrieval, but we use the correlation length from the version difference to propagate the uncertainty. We rewrite line 328-332 to make this more clear.

"To calculate the superobservation uncertainty resulting from the air mass factor uncertainty we use the uncertainty from the retrieval, together with a correlation c. Note that the AMF uncertainty is not part of the level 2 product, but can be calculated

using the available information provided in the ATBD (Geffen et al. 2022b). Calculating the associated spatial correlation of the AMF uncertainty is not trivial because the tropospheric air-mass factor $M_t$ is calculated through several inputs, algorithms, dependencies and feedbacks, as shown in figure 5."

Line 396-397: Is the green line just the average of the gray lines in Figure 8?

The green line is the standard deviation of the grey lines calculated using formula 16. We have added the following on line 398 to clarify:

"This standard deviation is plotted as the green line in figure 8a."

Line 421-423: It would be beneficial to further explain the calculation of fractional coverage (f), for example, by utilizing equations.

We have added the equation for fractional coverage in the appendix as equation B2 on line 705

Line 438-439: "At 50% coverage, the increase in RE is 54% for clean areas and 263% for polluted areas." Figure 8 does not support this statement. This sentence likely describes the inflation of representative error due to partially data coverage in Figure 9. Please consider moving this sentence to the right place.

This sentence indeed describes figure 9, which is an average of observations in time and space. Whereas figure 8 is only for a single superobservation. The sentence has been moved to line 462

Equation 18: This equation is provided but doesn't mention in the manuscript.

We now refer to this equation in the manuscript, fixing a latex labelling error that referred to the same equation in appendix B4.

Line 459: Could the author clarify what "sigma normalized RE" means here?

The RE varies significantly between superobservations depending on the internal standard deviation $\sigma$ of the grid cell. But after normalizing by the standard deviation all superobservations show similar values ($\sigma_{RE,n}$ /$\sigma$). This is further illustrated by formula 17

This is clarified by adding the following on line 406-408:

"Equation 17 indicates that $\sigma_{RE,n}$ is proportional to the standard deviation $\sigma$ of the observations within the grid cell. In figure 8 the results are divided by this grid-cell dependent standard deviation such that different TROPOMI superobservations can be compared."

Line 485-490: I am confused with these sentences. Did you mean that you first estimated $\sigma$ using Equation 16 and then computed the final representative error using Equation B5? Could author clarify on it?

> Because the RE has been normalized by the standard deviation up until this point, we now also need a good method of estimating the standard deviation. This section discusses this process. This is clarified in the previous comment.

Section 6: Many regional chemical data assimilation (DA) systems employ a finer horizontal grid (~ 10 km or less), which is comparable to the scale of a satellite pixel. In this scenario, the proposed method for estimating representative error may not be effective because the pixel population (N) becomes too small, leading to the use of the climatology value instead. I was wondering if the author could comment on the extent to which we should consider the representative error.

> This manuscript discusses superobservations that are larger than the satellite observation, prompting for a method to account for the different scales. When the scale/grid of the model or data assimilation system is similar to the size of a satellite footprint, the superobservation concept loses its validity, and individual observations should be used instead. But we have given the limit of N=1 explicit thought, and our approach is well behaved for small numbers of observations in a grid cell.

> Note that if the model has a higher resolution than the data there is also a representation error because the model grid is smaller than the observation. This error has not been considered in this research.

Line 494-495: Where did the intersection come from? I believe it is from Figure C1. Is that correct?

> This is correct. We have added this reference to line 478

> "Instead we set the standard deviation to 0.4 times the tropospheric column + 2.5 μmol m$^{-2}$. This is based on the relation between the standard deviation and tropospheric column as shown in figure C1 in appendix C."

Line 497: "uncertainty2 " Please fix the typo.

> We have rewritten this to $\sigma^2_{sob}$

Line 509: Please correct typo in this sentence for the figure citation.

> We have corrected the typo by adding a bracket.

Line 515-516: It is acceptable to retain all the descriptions here. I just want to comment on the fact that the random sample approach is not a very good method for data thinning given the large superobservation grid. A better way is to first analyze the distribution of all pixels inside a superobservation grid and then select one pixel that is closest to the mean or median. This method could pick up a more statistically representative data than a random sampling.

We introduced this simplest thinning approach purely as a reference to demonstrate what can be gained by using information from multiple observations.

We are aware of the approach to use the median of all observations within a grid box. Even though it seems only one observation is used, this approach actually makes use of all observations (to determine the mean or median), and we expect that this value will be quite close to our superobservation values. (However, one has to be a bit careful with medians when the PDF of values is asymmetric which may lead to differences between median and mean). Because information from multiple observations is used, the uncertainty of the mean or median will be smaller than the uncertainty of the single observation corresponding to the mean or median. A new uncertainty will have to be computed, following e.g. ingredients as discussed in our paper. If the original (too high) uncertainty is used, the assimilation of these thinned observations will be suboptimal because the analysis will not draw enough towards the observations.

We add some context on the usage of the median for thinning on line 78 of the introduction:

"A better approach is selecting the single observation closest to the mean or median of the observations within the gridcell (Plauchu et al., 2024), but note that this approach makes use of the information of all these observations."

Section 7.1: In light of the previous comment, it is not surprising that the thinning experiment demonstrated the worst performance in DA. A data-thinning experiment is too easy to outperform as a large discrepancy between model and superobservation is expected. Keeping this experiment in the manuscript is totally fine. I would like to encourage the authors to run one more experiment in which the superobservation (error) is just a simple average of all pixels' NO2 columns (reported error) that falls within the superobservation grid. Given that this approach is commonly used in many DA or data analysis studies, comparing the method proposed in this work to the "simple average approach" could help to highlight the benefit of the using advanced superobservation method. I recognize that it may be difficult for the author to run additional simulation, thus this is not required. Alternatively, it would be wonderful to see some discussion of the simple average method versus a more complicated superobservation method in the manuscript.

> The simple average approach is already present in the work in the form of the fully correlated experiment, with the minor difference of the average being done in variance space instead of uncertainty space. We have added a comment to line 530 to clarify this:
>
> "This is analogous to the variance averaged uncertainty."
>
> Furthermore, we have expanded the discussion of the superobservation and correlated/average experiment on line 570-579:
>
> "On the other hand, in the correlated case, the uncertainty is large, which reduced the data assimilation impact and somewhat increased the RMSE and MAD. This shows that assuming the uncertainties are fully correlated is not so unrealistic, but it does lead to a reduction in performance almost everywhere. One exception to this is Central Africa, where the lower uncertainty significantly improves the RMSE.
>
> Note that there is only a small decrease in the relative impact in this area going from the superobservations to the correlated experiment. Despite the fact there is almost no uncertainty reduction from the superobservations the uncertainty is still too low. It is likely that further increasing the uncertainty yields even better results than the correlated experiment.
>
> Because this effect is so strong and local we believe it is not related to the superobservation method, but instead results from fire related errors in the observation uncertainty or model. The high absolute errors in the area make for a large impact in the RMSE and MAD, despite a small difference in the relative impact. As a result the superobservations probably do not compare as good to the uncorrelated experiment in table 1 as they should"

Table 1: Why is the $\chi\chi$ 2 value for the uncorrelated error case (111.0) much larger than in other experiments? Could author verify this?

> This is indeed correct. The uncorrelated experiment has both very high errors due to overfitting and (too) low observational uncertainties. With the errors in the denominator and the uncertainties in the numerator this results in very high $\chi 2$ values

Table 1: The RMSE for fully correlated error and superobservation cases is similar. The $\chi\chi$ 2 value for the fully correlated error case is actually closer to one, which makes it difficult to conclusively say that the superobservation case outperform the fully correlated error experiment.

> Generally, the superobservations perform better than the correlated case as shown in figure 14d. The metrics are somewhat skewed due to high absolute errors in central Africa. The fire related emissions introduce a lot of variation, which is hard to capture with the model. Thus, one should not put too much weight one the results from this region.
>
> In this region the impact of the correlated and superobservation was similar, but still results in a large difference in RMSE. This makes us believe that both experiments underestimate the uncertainty related to either errors in the observational uncertainty or the model. The correlated experiments perform better in this region because the underestimation in the observational uncertainty is somewhat compensated not reducing the uncertainty. In any case the high absolute values in this region makes this error more pronounced in the metrics.

Line 583-584, Please change the "o-f" to "O-F".

> For clarity we have decided to use OmF instead of O-F, this has been applied everywhere in the text.

Section 7.1: The error budget analysis (Figure 11) shows that representative error is not the dominant source of the observation uncertainty. As the computation of representative error seems to be very costly, did the author have a chance to examine the impact of including or excluding representative error on the DA results?

> The RE is only relevant for observations with a substantially reduced coverage above the coverage threshold, which is a small part of all pixels. As a result, a majority of the superobservations have a small RE. However, the RE is very important for grid cells containing cloud edges. In practice we find that NO2 values near/at cloud edges show a larger spread indicative of sampling noise and larger errors. Such an increase in the uncertainty will reduce the impact of these observations in assimilation systems.
>
> Once the right parameters have been calculated the RE is not very costly to compute, and we do not propose to remove this term.

Section 7.1: I appreciate that the author conducted a thorough comparison between different DA experiments. Since the RMSE of many DA experiments are comparable, I would recommend finding an independent dataset for model evaluation to better understand the performance of individual experiments and determine which experiments better capture thereal NO2 fields. An observing system simulation experiment (OSSE) might also be useful in future work to test the performance of different superobservation techniques, since the ground truth is known.

> These are interesting ideas, but beyond the scope of this paper. In the future we plan to apply the superobservations in MOMO-Chem for creating a new reanalysis dataset. These results will then be compared with independent datasets, e.g. aircraft campaign measurements to investigate the degree of improvement obtained by assimilating the satellite data.

Line 600: Please define JAMES.

> This is the Japan Agency for Marine-Earth Science and Technology but the system is more accurately described as the MOMO-Chem DA system.

> On Line 601 we rewrite "JAMSTEC" to "MOMO-Chem"

Line 625: I am confusing about using correlation 0.15 for remaining observation error. This contradicts what is stated in Section 5.2, where the correlation is determined as a function of distance/correlation length, and each grid may have a different value for the AMF uncertainty. Could the author please clarify this?

> Here we are discussing another study, which applies different correlations. We clarify this by replacing "this work" on line 624 by "Sekiya et al. 2022".

Line 637: "According to Nyuist". It would be recommended to add a reference here.

> A reference has been added on line 637 to:

> "Shannon, C.: Communication in the Presence of Noise, Proceedings of the IRE, 37, 10–21, https://doi.org/10.1109/JRPROC.1949.232969,"

Line 640-644: The spatial correlation imposed on prior emissions and concentrations may be incorrect, and they are also updated by the DA process. As a result, using the predefined spatial correlation to justify the use of superobservations is not a good claim in my opinion.

Here we are discussing the correlations introduced by the DA system itself due to the inability of the system to resolve fine scale structures. Because the system is unable to resolve structures on the chemistry transport model resolution we hypothesize that it may be more representative to provide superobservation larger than the grid of the model.

Replace line 640-646 by:

"Data assimilation implementations typically introduce spatial correlation lengths covering multiple grid cells in the modeling of the background (forecast) covariance matrix **B**. These correlations act as low-pass filters and the fine-scale variability for smaller length scales is not constrained in the analysis. In that case, constructing superobservations larger than a single model grid cell could be explored, as long as the horizontal correlation lengths of the assimilation system are appropriately oversampled. These coarser superobservations could be useful for satellite data with a high relative noise level (e.g. HCHO and $SO_2$ column observations) or to reduce correlated uncertainties between observations while at the same time lowering computational costs."

Line 676: Please correct the typo in the equation.

This has been corrected

Equation A1: Please define $xx$, $ii$, $jj$, and n in the equation.

The definition of these variables has been added on line 687 as follows:

"for n number of tropospheric columns x in a superobservation indexed by i and j"

Line 712: The intercept in the figure is +2.5, not +3 mol m -2 written in the manuscript.

The number has been corrected from 3 to 2.5 mol m-2

---

## Author Comment (AC3)

**Comments Editor (DOI:10.5194/egusphere-2024-632-EC1)**

This paper thoroughly discussed and investigated constructing superobservation and the relevant uncertainty of the satellite NO2 data. However, it has serious drawbacks in writing and format. While addressing and responding to reviewers' comments and suggestions, I request that the authors also improve the manuscript to meet the standard of scientific writing. Please also see some other comments as follows.

We thank the editor for their feedback on further improving the writing and format of the paper. We have done some rewriting and restructuring to improve the readability of the paper. Below the answer to your comments. All line numbers in the answers refer to the new manuscript.

Line 73: Please briefly mention what are the disadvantages pointed out by Purser et al. (2000).

We have added the following on line 98-100:

"but Purser et al. (2000) points out two disadvantages with this method: Firstly the superobservations are not independent from the assimilation system and secondly, creating superobservations requires a statistical description of the forecast system, which is not always available"

Line 85: An unclear sentence.

We rewrite the sentence on line 111-112 to improve readability:

"Miyazaki et al. (2012a) and Boersma et al. (2016) average the observations with the overlap of the observation footprint with the superobservation grid as weights."

Line 87: (Inness et al. 2019b)

We have corrected this typo (line 115) to: "(Inness et al. 2019b)".

Line 165-172: Unclear description. Please revise these sentences with clear definitions. Is ym a simulated observation? Please clarify what will be used for obtaining x and xa.

We replace line 201-202 with the following text to improve clarity:

"Here x is the tropospheric $NO_2$ vertical profile from the model co-located in space and time to the footprint of the satellite and $x_a$ is the a-priori vertical profile used in the retrieval."

Line 176: Please use a formal style for the section title.

This section has been removed due to restructuring, thus the section title is removed.

Line 196: I can't follow the argument that "given the same uncertainty, low NO2 observation force assimilation more than high observations". Did you mean low NO2 observation will have small uncertainty if the uncertainty (measured in percentage) is proportional to the column amount?

This section is removed due to the restructuring. The content can now be found in the introduction on line 85:

"If all individual observations with their individual uncertainties are assimilated in a model with a coarser resolution than the satellite, this leads to low-biased analyses, because more weight is given to low observations with a small uncertainty. With the superobservation approach described in this paper, such persistent low biases are largely avoided."

How to derive the weights shown in Fig. 2?

The weights are the area overlap between the superobservation grid cell and the satellite footprint. We add the following to the description of the figure 2 to clarify:

"The colours indicate the weight $w_i$, which is the area overlap ($km^2$) between the superobservation grid-cell and satellite observation footprint."

Figure 3 is shown, but I can't find the relevant discussion.

We add a reference to the figure on line 220:

"An example of this method is shown in figure 3."

It is not clear why Eqs. (8) and (9) are discussed. Did the authors want to explain how the superobservation affect the calculation of innovation?

For properly comparing (super)observations to a model you need to apply the satellite kernel to the model. Thus making superobservations not only requires calculating a representative observation, but also the corresponding averaging kernel. We make this more clear by adding the following in line 232:

"To compare superobservations against a model we also need a corresponding averaging kernel, which are averaged in the same way as the observations. Multiplying Eq. 1 with $w_i$ and summing over the satellite observations we get:"

(11): Please clarify how to obtain the correlation factor, c. What is the value of c in this study?

The correlation factor c depends on the type of uncertainty. It set to 1 for the stratospheric uncertainty, set to 0 for the slant column uncertainty and defined by a correlation length for the AMF uncertainty. We make this more clear by changing lines 265-268 to:

"As mentioned in section 3.2, the superobservation uncertainty depends on the observational uncertainties and their correlation c (Eq.11). As shown in equation 2 the tropospheric column uncertainty consists of 3 separate sources of uncertainty: The stratospheric uncertainty, the slant column uncertainty and the air mass factor uncertainty. The superobservation uncertainty of these components is calculated separately because they have different correlations, which means their uncertainty propagates differently. Every component and its correlation is discussed individually in the sections below. "

Line 532-540: Section 7.1 needs significant revision. The experiment configurations should be provided with clear descriptions.

We add extra information on the experiment configurations in line 516:

"The assimilation is run with 32 ensemble member and an assimilation window of 2 hours. The localizations are based on a species-dependent localisation scale. These are derived from sensitive tests in Miyazaki et al. (2012b). Covariance inflation is achieved through the inflation of emission factor uncertainties, by inflating the spread to a minimum predefined value. Additionaly, a multiplicative covariance inflation of 7% is applied to the concentrations. The details of the assimilation approach that is used are described in Miyazaki et al. (2020b). In addition to $NO_2$, we also assimilate total column CO from the TIR/NIR band of the Measurement of Pollution in the Troposphere instrument. (MOPITT) (Deeter et al., 2017), the $SO_2$ planetary boundary layer vertical column from OMI (Li et al., 2020), and Aura Microwave Limb Sounder (MLS) $O_3$ and $HNO_3$ profiles (Livesey et al., 2022). To demonstrate the impact of different superobservation settings, the following 4 sensitivity runs were done for July 2019, only varying the $NO_2$ observations:"

---

## Author Response (AR2)

**Reply Topic editor decision:**

As both reviewers gave very positive comments on this manuscript, I suggest that the manuscript be accepted after the reviewers' and my minor comments/suggestions are adequately addressed.

We once again thank the editor for their feedback. We are glad our changes have been well received. Below we address the new comments.

Fig 3 and Fig. 12a, d are the same. Since Figure 3 is only used for an example with little explanation, there is no need to duplicate the plots.

We have removed figure 3. The references in the text now point to the other figure. (now figure 11a and 11b). The text has also been slightly to accommodate this:

"Figure 11b shows satellite observations over Europe, with the associated superobservations in figure 11a"

The air mass factor uncertainty is the only one with a horizontal correlation pattern. Referring to Fig. A2b and mentioning the correlation length (32 km) in section 4.3 would be clearer for the readers.

We have added the correlation length on line 362:

"Here we find a correlation length of 32 kilometres."

However we believe that we should not refer to figure A2b from here, because the main text does not contain enough information to properly interpret the figure. A reader that would jump from section 4.3 to Fig. A2b and back will probably misunderstand the figure and get a wrong idea on the method we use to calculate the correlation length.

We do add an approximate correlation for the air mass factor that results from this correlation length for a 0.5 degree superobservation on line 375:

"For a 0.5 degree superobservation this gives $C \approx 0.3$"

Please clarify what you meant by "the observation uncertainty is fully correlated in space" for the data assimilation experiments. c=1 for all three uncertainties and RE?

Here we mean that the correlation C from equation 11 is equal to 1 (C=1). We have added the correlation values for the uncorrelated experiment on line 528:

"The standard superobservations, with modified uncertainty assuming that the observations are fully uncorrelated in space (C=0, fig. 12d)."

And for the fully correlated experiment on line 530

"The standard superobservations with modified uncertainty assuming that the individual observations are fully correlated in space (C=1, fig. 12b)."

Also we add these values to the description of figure 12:

"Figure 12. Panel showing various methods of pre-processing uncertainties for data assimilation and the RE on 2018-09-08 for qa > 0.75. (a) superobservation uncertainty constructed for this research (b) fully correlated uncertainty (C=1) (c) representation error (d) uncorrelated uncertainty (C=0)"

Please clarify what is used for O_t,x,y in the RMSE calculation since four experiments use different superobservations.

On line 540 we clarify this variable:

"Here $O_{t,x,y}$ are the observations associated with the experiment (1, 2, 3: fig. 11a and 4: fig. 11d) and $F_{t,x,y}$ are the forecasted values. This is shown in figure 13."

**Referee #1 report:**

I would like to thank the authors for greatly improving the manuscript. The paper seems ready for publication.

We thank the referee for reading our revised manuscript. We are glad the changes were well received

**Referee #2 report:**

The authors had thoroughly addressed my comments and questions. The revised manuscript has significantly improved its readability and content compared to the previous version. Excellent work!

I only found a few typos in the revised manuscript that could be addressed during the proofreading stage:

> We thank the referee for reading the manuscript, we are glad the changes were well received. Also we thank you for your thorough inspection of the revised manuscript and catching these mistakes. We address them below:

Line 405: There was no equation 8a in the manuscript. I think the authors are referring to equations 16 or 17.

> Thank you for catching this mistake, we now refer to equation 17.

Line 481: Please consider changing the symbol of superobservation uncertainty from $(\sigma\_sob)^2$ to $(\sigma\_s)^2$ in alignment with the definition in equation 10.

> The symbol should be consistent. We have changed line 481 to:

> "Figure 10 shows the contributions to the superobservation uncertainty ($\sigma^2_s$ ) as a function of the tropospheric NO2 column."

Line 598: The HARP toolbox first appeared in Section 3.1; please consider defining the acronym in this section.

> We now define the HARP acronym on line 221:

> "This method of averaging is similar to spatial binning using the Data harmonization toolset for scientific earth observation data (HARP)"

> Line 600 now reads:

> "This is similar to the method of Inness et al. (2019b) and the HARP spatial binning method for total uncertainty variables"

**Changes to colour schemes:**

We have updated the colour schemes on the following figures to make them more readable for readers with colour vision deficiencies:

- Figure 10
- Figure 11
- Figure 12
- Figure 13a
- Figure 14a
- Figure A2 b, c and d
- Figure C1